# Neighborhood Identity Formation and the Changes in an Urban Regeneration Neighborhood in Gwangju, Korea

Hae Young Yun [1],* and Hyun-ah Kwon [2],*

1  Asia Research Institute, National University of Singapore, Singapore 119260, Singapore
2  Department of Architecture, Mokpo National University, Muan-gun 58554, Republic of Korea
*  Correspondence: yunxx051@umn.edu (H.Y.Y.); helenack@naver.com (H.-a.K.)

**Abstract:** Since the Urban Regeneration Act in 2013, central and local Korean governments have endeavored to regenerate deprived urban neighborhoods. This study analyzed how these efforts have changed the nature of neighborhood identity in Yanglim, Gwangju, Korea. The authors analyzed 62,386 Naver blog posts from 2013 to 2022, utilizing an Artificial Intelligence (AI) technique, Topic Modeling (i.e., Latent Dirichlet Allocation). Using trend analysis by topic, three phases were identified: (1) Phase 1: Flourishment (January 2013 to October 2016); (2) Phase 2: Maturation (November 2016 to February 2020); and (3) Phase 3: COVID-19 (March 2020 to October 2022). In the first phase, the collective actions between the local government and citizens to improve the declined neighborhood formed the Yanglim area's reputation as the "History and Cultural Village" and as "Penguin Village". The unique identity of the area in the second phase, along with gentrification issues, created a hot spot (e.g., cafés and restaurants), drawing the attention of tourists and locals. More recently, the Yanglim area has become a place for locals' daily activities with their loved ones, as tourist traffic greatly dropped off due to the COVID-19 outbreak. Until now, the Yanglim area has experienced a process of successful urban regeneration from flourishment to degentrification. AI techniques represent a novel application that can support policy makers and stakeholders in understanding citizens and taking further actions to create economically and socially sustainable neighborhoods.

**Keywords:** urban regeneration; neighborhood identity; identity change; history and culture; hot spot; COVID-19

## 1. Introduction

The identities of urban spaces or neighborhoods are manifested by their people (e.g., their residents and visitors). These identities are shaped by unique images, not only due to landscapes and buildings but also due to food, music, customs, language, relationships, and political viewpoints. All of these aspects (i.e., physical and cultural) interact with each other to form unique identities [1]. Lynch defines identity as "the extent to which a person can recognize or recall a place as being distinct from other places" [2] (p. 131). Oktay states that "identity is one of the essential goals for the future of good environments. People should feel that some part of the environment belongs to them individually and collectively, some part for which they care and are responsible, whether they own it or not" [3] (p. 261). As stated above, human beings are important stakeholders interacting with built environments. Thus, it is important to research human perceptions and their interactions, identifying the distinctiveness of different neighborhoods.

In recent decades, relevant stakeholders (e.g., local governments, urban planners, community leaders and residents) have been challenged to revitalize dilapidated neighborhood environments as well as to mitigate negative images by collective actions in many countries. The proliferation of collective actions is categorized as "commercial-led regeneration", "culture-led regeneration", "tourism-led regeneration", "design-led regeneration",

or complexes of these terms [4–7]. As one of the countries implementing urban regeneration policies with a proliferation of projects, South Korea (hereafter, Korea) has worked to revitalize depopulated and economically unsustainable city centers. Accordingly, a number of articles dealing with urban regeneration have been published in Korea; the majority of research has shed light on Seoul and the metropolitan areas, with limited knowledge of other cities such as Gwangju.

In recent years, utilizing social media platforms to mine people's opinions in order to broaden ideas on urban environments has become widespread. In particular, Natural Language Processing (NLP) and Artificial Intelligence (AI) techniques are applied to mine the opinions of diverse people, overcoming the usual time and budget limitations for data collection [8]. Using big data analysis has great significance. First of all, an overview of social media permits new and chronological trends to be uncovered. Second, this paper illustrates the potential of topic modeling to the greatest extent [9]. This technique can offer invaluable insights to relevant stakeholders such as policy makers and planners for the purpose of understanding citizens, which is necessary for urban regeneration. Nonetheless, a great many previous studies have concentrated on qualitative and phenomenological approaches such as interviews, observations, or mapping to reveal urban or neighborhood identities.

Capitalizing on the benefits of social media data, the authors aim to answer the two following questions:

1. Do social media data allow us to capture a certain point in the changing nature of neighborhood identity through AI techniques?
2. If so, is it possible to capture how changed the neighborhood identity is at the discursive level over time?

To answer these questions, the authors utilized the lens of neighborhood stakeholders and users of the space, including locals, small business owners, tourists, and government officials who posted their opinions and activities on social media during a certain period of time and then analyzed these data with a newly emerging computational method.

## 2. Literature Review

### 2.1. Understanding and Measuring Neighborhood Identity

In this section, the authors review how research trends on neighborhood identity have changed. Lynch (1960) addressed the image of a city and how urban planners can create memorable city images in the book The Image of the City [10]. The image of a neighborhood space is constructed in a two-way process between human beings and their environment as people select, organize, and endow what they see with meaning. This image differs from person to person, and different neighborhood environments are associated with different processes of image-making. An environmental image may possess three components: identity, structure, and meaning. Among these three components, identity implies distinctiveness compared to others and its recognition separate from another entity. A neighborhood identity is a collective representation created by subjective perceptions of any single individual or group rather than by objective reality in the neighborhood [11–13]. Understanding neighborhood identity based on Lynch's [10] and Suttles' work [14], researchers have found that people interact with their neighborhood as a creative imposition and that people's activities are connected to their 'cognitive maps'(i.e., their internal representation of the space) [15].

In one stream of urban research, researchers have mainly studied how urban or neighborhood identity is depicted using two methodological approaches: (1) traditional methods, and (2) computational techniques. The first addresses the interdependent identity as perceptions among external observers (e.g., tourists, non-natives), the in-group (e.g., natives, residents rooted in place), and stakeholders (e.g., policy makers) [3,11,16–20]. Their perceptions are measured using traditional methods (e.g., interviews, survey questionnaires, diaries). Moreover, researchers have added their own perspectives through observations. For example, Huovinen and colleagues tried to measure neighborhood identity by

using interviews and diaries to gauge residents' perceptions [20]. Salesses and colleagues created a discursive map with different key words in four different places, then related the identity of places to different perceptions [21]. In addition to methodologies such as interviews and survey questionnaires, the systematic reconstruction of cognitive maps of neighborhood boundaries by relying on residents' perceptions represents a methodological challenge. The boundaries of neighborhoods should be decided by the average or the largest example [22]. In addition to being expensive, these methods are difficult to validate [23] and do not capture enough of the meanings and identities of neighborhoods that are attached to languages [13].

The second approach uses computational techniques to gather and process high volumes of data through Geographical Information Systems (GIS), space syntax, and programming languages (e.g., R and Python) [1,21,24,25]. For example, geo-tagged digital images collected from four different places through online maps and on-site observations were analyzed in the context of city identity. The safety, uniqueness, and social class of neighborhoods were used as a proxy of city identity as measured by the perceptions of the public (i.e., crowdsourcing). The results were then be visualized as maps. Zhou and colleagues analyzed over two million geo-tagged photos from Google Street Views, Flikers, and Panoramio to identify city identities from twenty-one different cities [25]. Their study labeled scene attributes with one hundred and two distinctive classifications, such as natural, eating, and open areas by drawing on deep learning processes. Later, spatial analysis was performed with these characteristics on maps and a similarity network analysis was created.

In addition to image processing for place identity, determining the linguistic patterns in big data has not yet fully revealed its potential. Research has been performed with big data, such as that from Twitter, TripAdviser, Yelp, and news article data from diverse geographical regions [1,8,13,26]. Such works have explored, for instance, (1) how similar or changed neighborhoods are over time [13], (2) how different city characteristics or urban green space characteristics are from each other [1,8], and (3) how different the topics are based on geo-tagged locations [26]. Certain data types, such as from TripAdviser and Yelp, only provide insights for commercial destinations and do not cover diverse opinions. Although Twitter covers spatiotemporal information with texts being used by diverse populations, the number of users in Korea is limited and the data do not cover a long-term period. Thus, the authors mainly focused on an analysis of how neighborhood identities have been constructed through an emerging methodology by analyzing big data (i.e., Naver blogs), thereby adding more knowledge to the literature. Through big data analysis, the authors adopted text mining to deal with linguistic patterns for neighborhood changes using data accumulated over a period of almost ten years.

### 2.2. Understanding the Study Neighborhood

Urban decline is a multidimensional process describing the reduction of local economic opportunities along with a decrease in jobs and a rise in unemployment, depopulation, and dilapidation of the neighborhood environment [27]. Central and local governments in many developed countries have tried to lessen the negative effects of urban and neighborhood shrinkage and to bring back the positive image of cities and neighborhoods. Korea is one of the countries investing tremendous amounts on urban regeneration. As an exemplar of local cities, the Yanglim neighborhood in Gwangu was chosen for this study. Gwangju is the sixth largest city located in southwestern Korea. It covers 501 km$^2$ and had almost 1.5 million people in 2021 [28]. It is one of the cities currently experiencing a gentle population decline [29].

Yanglim is a residential area covering 0.68 km$^2$ and surrounded by the Sajik and Yanglim mountains in Gwangju. The Gwangju stream is located nearby. While Yanglim is located close to the old city center, prior to 1904 the land around it was affordable, and included a great many graveyards. Taking advantage of affordable land, missionaries from the Presbyterian church in the United States settled in the area, using it as a base camp

for their religious missions and social work [30]. Their most outstanding activities were education, medicine, and mission, establishing schools (e.g., Speer Girls' Schools, the first institute for girls' education in Gwangju), a hospital (the current Gwangju Christian Hospital), and a church (Yanglim Presbyterian Church). The Yanglim area played a role as a center for the arts (e.g., traditional Korean and Western music and performance), intellectual exchange, and the movement for independence from Japan and was a center for religion, medicine, education, and social work. Until 1970, this place was called a "western village", "Christian village", or "cradle for modern culture in Gwangju", with a modern historical heritage and Korean traditional houses (e.g., western-style buildings and Hanoks). In addition, the Yanglim neighborhood is the home town of famous artists [31]. Since 1976, the lower part of Yanglim has been developed as a residential area to solve the housing shortage; most of the housing in this neighborhood is 30 years old or more, with some vacant and dilapidated houses. The aged houses, narrow streets, public health issues such as streets inaccessible to fire trucks, and lack of parking lots and walking paths have been considered important issues for improvement [32].

In 2009, the local government launched initiatives to develop the Yanglim neighborhood as a historical and cultural village to attract tourists [33]. The local district assigned funding (approximately USD 24 million) to regenerate the neighborhood over a nine-year period (2009–2017). The first initiative was the renovation of the missionary heritage, the formation of a missionary memorial park, and the maintenance of walking paths in the neighborhood. This initiative included the construction of a memorial tower (the current Sajik Observatory Tower) along with renovations of old schools and the medical center [32]. During this period, the local government focused on developing art and exhibition content and constructing several small art galleries [34]. In addition to these local government initiatives, the residents voluntarily began beautifying the neighborhood. One resident started to clean up a burnt-down house, and hung items such as old-style clocks on fences and walls for decoration. Other residents began urban farming in the vacant lots, accelerating the urban regeneration process. The collective actions of the residents, the local artists, and the local government have made the neighborhood, now known as Penguin Village, widely renowned. "Penguin" is the nickname of a resident who waddles like a penguin after a car accident, and who has been actively involved in the neighborhood project [35]. Penguin Village has become renowned, drawing a large volume of tourists and locals. This has functioned as a driver of neighborhood change, stimulating construction and renovation of commercial buildings and houses to suit the demands of both insiders and outsiders. On top of these nine years of improvements, the local government secured additional funding (approximately USD 51.6 million) for the next six-year period (2018–2023). The goals in this period are as follows: (1) improvement of the residential area (e.g., funding and program support for housing repairs and securement of parking spaces); (2) improvement of the urban environment (e.g., creation of a smart city and Book Street); (3) vitalization of the local economy (e.g., by attracting small business owners and opening an urban regeneration support center); and (4) social integration (e.g., the management of a local community and citizen art school) [36]. To realize these goals, the local government has incentivized local artists and small business owners to settle in the area by subsidizing rent and offering affordable studios in cooperation with the neighborhood community. These endeavors have included constructing new buildings and launching additional programs for the local economy, arts, and community through incentives and sponsorships.

## 3. Methodology

This section describes the methodology used to detect neighborhood identity and the changing nature of the Yanglim neighborhood in Gwangju. Three stages were set up for the data collection and analysis process using computational technique and statistics analysis. The entire process is shown in Figure 1. It is possible to draw out individuals' impressions and behavioral patterns from the analysis of blog data. Thus, in the first stage, we collected data from the Naver website for a ten-year period. In the second stage, after

data cleaning, topic modeling and trend analysis with all data were performed in order to explore the neighborhood identity. In this stage, a great volume of unstructured text data were converted into quantifiable data. In the third stage, after dividing the dataset using the trend analysis, the authors analyzed how the neighborhood identity changed over the study period. The detailed methodology is described below.

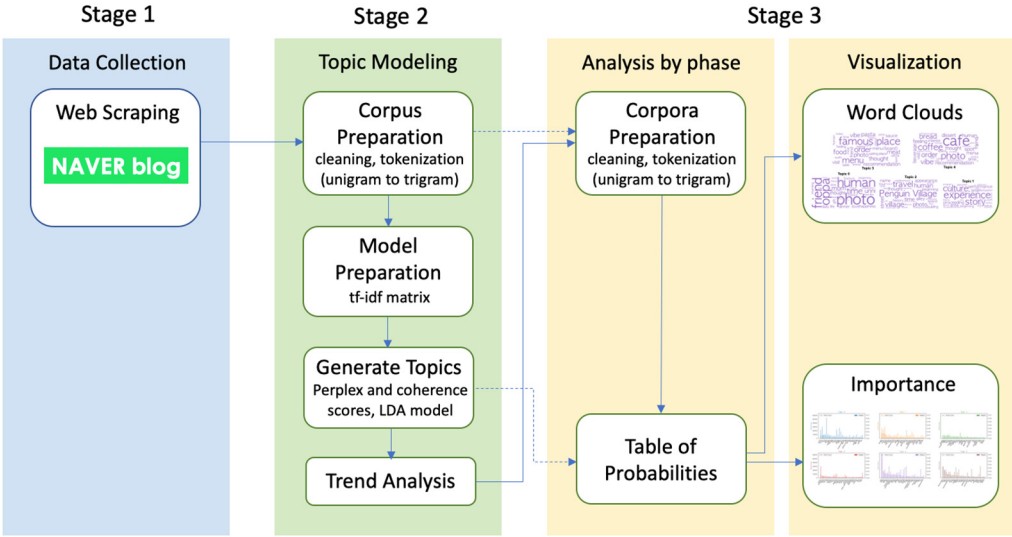

**Figure 1.** Research framework for this research.

### 3.1. Stage 1: Data Collection

Web scraping is a type of data mining that involves extracting unstructured data from targeted websites, then transforming these data into structured data and storing them as a file or in a database [37]. Using this data collection technique, the keyword, Yanglim-dong was searched on the website Naver Blogs for a ten-year period from January 2013 to October 2022. Naver is one of the largest portals in Korea, and has provided a blog service since October 2003. Thus, the accumulated amount of data is quite massive. The blog dataset includes information about what happens in the area, how and where users are, as well as what their opinions are. The voices and behaviors of users include tourists, visitors, residents, local business owners, policy makers, and tourism stakeholders; these data can be used to gain information on users' diverse activities and opinions. Geo-tagged information was collected, if available, in order to determine whether the postings were about Yanglim. The dataset, consisting of 96,356 blog posts in total, was scraped through self-implemented Python programming by the first author, and a total of 62,386 blog posts were left for analysis after the data cleaning process. Data cleaning was performed using keywords (e.g., rental car, potentially tagging various neighborhoods beyond Yanglim) as well as with geotagged information, blog user IDs, and even reading of specific blog lines. Certain bloggers tagged Yanglim or Yanglim-dong in order to make their blogs highly searchable, even when the postings were not relevant to Yanglim at all. Thus, the data cleaning process was important to securing a fine dataset.

### 3.2. Stage 2: Topic Modeling for All Data

The topic modeling used Latent Dirichlet Allocation (LDA) calculated for the data analysis, which is an unsupervised method [38] using an Artificial Intelligence (AI) technique. Therefore, all collected data needed to be written in one language (i.e., Korean). This section illustrates the corpus preparation, model preparation, and topic generation processes. A text corpus is defined as a large and unstructured set of text. Recently, it has been defined as a set of data which is electronically processed and stored for statistical analysis and hypothesis testing that confirms occurrences or validates linguistic rules within a certain language territory [39]. "A corpus does not contain new information about

language, but the software offers us a new perspective on the familiar. In order to gain this new perspective, the first analytical steps generally involve two related processes: the production of frequency lists (either in rank order, or sorted alphabetically) and the generation of concordances" [40] (p. 122). In corpus preparation, tokenization is the very first step in text processing. Korean is an agglutinative language, meaning that combinations of nouns with or without whitespaces generate the same meaning. This leads to difficulties in tokenization. MeCab in KoNLPy, a Python library for NLP, was originally developed for Japanese morpheme analysis, and has been modified to MeCab-Ko for Korean morpheme analysis [41]. Compared to other libraries, MeCab-Ko provides stronger tokenization in terms of both speed and performance. A major strength of MeCab-Ko is that it offers users the function of an editable library of proper nouns. The blog contents that the first author collected contained a great many proper nouns (e.g., human or business names) and loanwords or English words directly written in Korean. Combinations including these words can result in less accurate tokenization. Thus, the authors input a proper noun list into the library of MeCab-Ko after reviewing the collected blog contents. To improve the quality of the tokenization and to perform better topic modeling, the raw corpus then needed to be cleaned of special characters, URLs, punctuation, and stopwords which are less significant in semantic contexts. Tokenization was performed by morpheme, then only nouns were selected into the tokenized list. Items from the tokenized list with fewer than three corpora were dropped as well.

This study implemented LDA, "a generative probabilistic model of collections of discrete data such as text corpora" [38] (p. 994), to analyze topics. Model preparation started with vectorization of documents from the corpus. To prepare the model, the vectorization process assigned a unique identification number for each word. While the unigram corpora (individual words) do not exactly capture the meaning of two to three words in a sequence, the n-gram analysis made of n words benefits from the co-occurrence observations [42]. By applying the bigram_mode and trigram_mode phrase modeling models in turn, the frequency of the corpus was counted and the result was transformed into a dictionary with the ID as the query key. The tokenized list combined with unigram to trigram was saved into a dictionary and the words or terms from documents were transformed into tuples (i.e., from document to bag-of-words) for their subsequent numerical calculation.

To generate topic modeling, a Term Frequency–Inverse Document Frequency (TF-IDF) matrix was used to compute the frequency of the terms in a corpus of documents (i.e., term X appears in document Y) [43]. LDA starts from the intuition that documents exhibit multiple topics [44], and is a part of the larger field of probabilistic modeling. In generative probabilistic modeling, researchers deal with data arising from a generative process including hidden variables. A joint probability distribution over the observed and hidden random variables is defined by this generative process. Data analysis is performed by harnessing the joint distribution to calculate the conditional distribution, which is called the posterior distribution, of the hidden variables with the observed variables [44]. Under the given requirement, the LDA model groups documents with at least two relevant topics [45]. The LDA model is one of the most important and widely used probabilistic models [46].

Because the number of topics k is a significant parameter for topic modeling, k was determined by the calculation of coherence and perplexity. On the basis of these scores, the better model (i.e., higher value) for the desired topic number was chosen. The authors acquired up to the top thirty most relevant terms from each topic and created visualizations of those terms using word clouds and graphs based on the relative importance of the topic. In particular, the visualization of word clouds aimed to represent the percentage of individual topics that emerged, with the size of individual words standing for the importance within each topic, not across all topics. The importance of words for the LDA was extracted using a document–word matrix $wd[w, f_d]$ indicating the importance of a word w in document $f_d$; please refer to the work of Maskeri and colleagues for further details [47].

Whole-topic saliency was computed based on the following formula:

$$\text{Distinctiveness(w)} = \sum_{T} P(T|w) \, log \frac{P(T|w)}{P(T)}$$

"For a given word w, we computed its conditional probability P(T|w): the likelihood that observed word w was generated by latent topic T. We computed the marginal probability P(T): the likelihood that any randomly selected word w' was generated by topic T. We defined the distinctiveness of a word as the Kullback–Leibler divergence [48] between P(T|w) and P(T)" [49] (p.2). For more detail, please refer to the work of Chuang and colleagues [49].

Labeling topics and thematic analysis of topics are interactive processes that involve human interpretation, although topic modeling itself is an automated process [50]. The grouping of each topic (i.e., thematic analysis) depends on the results of the Intertopic Distance Map used to determine the topic distances through an automated process. Thus, the authors decided on the labels of the topics and themes considering the keywords and the Intertopic Distance Map. The percentage of each topic was calculated during the generation of the Intertopic Distance Map.

*3.3. Stage 3: Topic Changes by Period*

The government interventions have been ongoing since 2009, and the interventions during the data collection period were not likely to have had any immediate effects on the responses of the locals and tourists posting on social media. Thus, trend analysis by topic was performed to determine how to categorize a period by topic change instead of dividing the period by government interventions. First of all, the LDA model assumes each document contains more than one topic, as mentioned above. Thus, the percentage contribution by each topic for each document was computed throughout all the documents. After that, each assigned topic was computed by year and month (e.g., January 2013).

Next, the authors used the same procedures for cleaning, tokenization, and bag-of-words (e.g., trigram modeling) detailed in Section 3.2. Because the rate of each topic presented differed by phase, the number of topics and the contents of the topics that emerged were inconsistent for each stage. As described above, the authors computed the perplexity and coherence scores, then selected the better LDA models based on these values. The results were drawn as word clouds and the word counts and importance of topic keywords were illustrated in figures. The individual topics were labeled considering the themes of the keywords, as described above. For instance, the topic containing keywords such as *travel*, *Penguin Village*, and *alleyway* was labeled as tour and culture. The saliency of terms for each phase was computed using the formula stated above.

**4. Results**

*4.1. Topic Modeling for All Data*

As described above, all data (January 2013 to October 2022) were analyzed in this stage to detect neighborhood identity for the entire period of time. On the whole, the exclusive identity of Yanglim, as a hot spot and tourist destination for sightseeing, consists of famous cafés, restaurants, and Penguin Village. The most salient terms that emerged were *café* (96,038), *photo* (79,606), *famous place* (54,285), *time* (44,234), *menu* (42,083), *order* (41,758), *coffee* (32,203), *vibe* (31,971), *recommendation* (29,206), and *feeling* (25,778). Other terms listed in this analysis are *travel, Penguin Village, space, food, unni* (which means older females of similar age or in a blood relationship in Korean), *mom, dessert, bread, beverage, looking around, table, pasta, meat, village, sauce, street, interior, alley, tour, and culture* (Table 1).

**Table 1.** Most salient terms and their frequencies by stages.

| All | | Stage 1 | | Stage 2 | | Stage 3 | |
|---|---|---|---|---|---|---|---|
| Hot Spot and Tourist Destination | | Cultural Heritages and Tourism | | Hot Spot and Cultural Tourism | | Daily Life and Hot Spot with Loved Ones | |
| Keywords | Frequency | Keywords | Frequency | Keywords | Frequency | Keywords | Frequency |
| café | 96,038 | photo | 4927 | café | 37,563 | café | 53,767 |
| photo | 79,606 | human | 4673 | photo | 31,725 | famous place | 29,929 |
| famous place | 54,285 | travel | 3304 | famous place | 20,894 | order | 22,162 |
| time | 44,234 | thought | 3179 | menu | 18,836 | menu | 21,439 |
| menu | 42,083 | culture | 2572 | order | 18,623 | coffee | 17,525 |
| order | 41,758 | Penguin Village | 2550 | friend | 14,871 | recommendation | 16,528 |
| coffee | 32,203 | café | 2289 | vibe | 14,224 | vibe | 15,644 |
| vibe | 31,971 | village | 2150 | coffee | 12,204 | visit | 15,193 |
| recommendation | 29,206 | story | 2049 | recommendation | 11,334 | feeling | 14,186 |
| feeling | 25,778 | area | 1833 | travel | 11,028 | spot | 13,719 |
| travel | 23,915 | space | 1763 | feeling | 10,209 | space | 11,321 |
| Penguin Village | 23,385 | friend | 1671 | visit | 9561 | unni | 11,066 |
| space | 20,242 | artwork | 1274 | food | 8732 | mom | 10,753 |
| food | 19,521 | alley | 1263 | bread | 7552 | dessert | 10,391 |
| unni | 18,183 | proceeding | 1212 | table | 6318 | food | 10,111 |
| mom | 17,958 | looking around | 1173 | dessert | 6148 | oppa | 9090 |
| dessert | 16,834 | Korea | 1160 | pasta | 5917 | travel | 8643 |
| bread | 16,125 | coffee | 1103 | village | 5904 | bread | 8288 |
| beverage | 14,228 | art | 1035 | beverage | 5768 | beverage | 7969 |
| looking around | 14,143 | Mr. | 1003 | meat | 5285 | table | 7053 |
| table | 13,809 | experience | 993 | menu(board) | 5108 | Penguin Village | 6901 |
| pasta | 12,878 | missionary | 941 | oppa | 4852 | pasta | 6251 |
| meat | 11,769 | church | 863 | sauce | 4772 | meat | 6181 |
| village | 11,523 | artist | 856 | eating place | 4451 | sauce | 6059 |
| sauce | 10,743 | order | 799 | cake | 3567 | pizza | 5615 |
| street | 9816 | menu | 690 | pizza | 3393 | artwork | 4725 |
| interior | 9537 | movie | 619 | sushi | 2861 | exhibition | 3736 |
| alley | 7942 | China | 528 | steak | 2349 | village | 3662 |
| tour | 6929 | (musical) performance | 179 | Testa | 2277 | artist | 3235 |
| culture | 6675 | music | 166 | experience | 2222 | soup | 2610 |

As explained in the methodology section, the authors calculated coherence and perplexity scores to identify better models with an optimal number of topics; in the end, an LDA model with five topics for all data was calculated. Based on the LDA modeling, word clouds and the count and importance of keywords were generated. Figure 2 illustrates the top thirty keywords composing the five selected topics out of the entire set of postings. During the entire period, the emerging topics were as follows: Topic 0 (daily life: 33.2%), Topic 1 (tour and culture; 10.3%), Topic 2 (café and hot spot; 17.5%), Topic 3 (food and hot spot; 18.6%), and Topic 4 (culture and citizen participation; 20.4%; Appendices A and B).

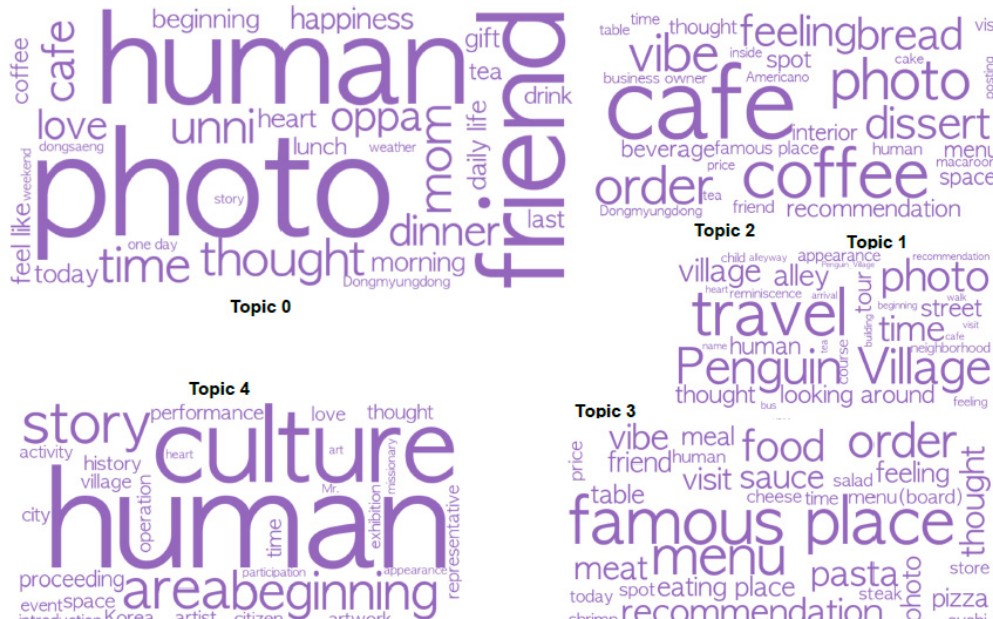

**Figure 2.** Word clouds for all data.

Topic 0 is labeled daily life, which includes people or loved ones interacting in daily life. The keywords of importance for Topic 0 are as follows: *photo* (0.0153), *human* (0.0109), *friend* (0.0095), *thought* (0.0080), *time* (0.0077), *mom* (0.0075), *unni* (0.0073), *café* (0.0060), *oppa* (0.0057; opposite gender of *unni* in Korean), and *dinner* (0.0056). Topic 1 (tour and culture) is related to historic and cultural resources present or cultivated in the Yanglim area. Topic 1 incorporates keywords such as *travel* (0.0326), *Penguin Village* (0.0278), *photo* (0.0225), *time* (0.0139), *village* (0.0109), *alley* (0.0088), *looking around* (0.0087), *street* (0.0086), *thought* (0.0085), and *human* (0.0082), in order of importance. Topic 3 is about cafés and hot spots. This topic contains keywords including *café* (0.0684), *coffee* (0.0244), *photo* (0.0165), *vibe* (0.0137), *order* (0.0137), *dessert* (0.0096), *bread* (0.0092), *feeling* (0.0089), *recommendation* (0.0087), and *spot* (0.0085). Coffee and dessert-related keywords occur as well. Topic 3 is about food and hot spots. The keywords (e.g., *famous place*, *menu*, and *order*) are concurrent with Topic 2, but food-related terms are dominant here. Topic 4 represents culture and citizen participation; the relevant terms are illustrated in Figure 3.

The resulting trends of the topics are shown in Figure 4. In particular, the rate of Topic 4 (i.e., culture and citizen participation) is noticeably higher than others before 2017. Topic 1, tour and culture related to Penguin Village, shows a similar pattern to Topic 4 since 2015. However, other topics are comparable to each other in the first part of the graph, showing only moderate increases or decreases in rate. Moreover, the rate of Topic 4 dramatically dropped after the COVID-19 outbreak in Gwangju. Thus, the authors separated the periods of analysis into three: (1) January 2013 to October 2016 (stage 1: flourishment); (2) November 2016 to February 2020 (stage 2: maturation); and (3) March 2020 to October 2022 (stage 3: COVID-19). The numbers of documents contained in each stage are as follows: 5362 for stage 1, 26,408 for stage 2, and 30,616 for stage 3.

Figure 5 presents the topic counts by documents and the five major terms generated through LDA modeling. As seen in Figure 5, the volume of topics related to culture and citizen participation was higher in the first stage than for others. Over time, the volumes of other topics balanced one another, then the daily life-related topic (Topic 0) dramatically outweighed the others during the COVID-19 period (stage 3). Overall, the quantity of blog postings increased noticeably over time.

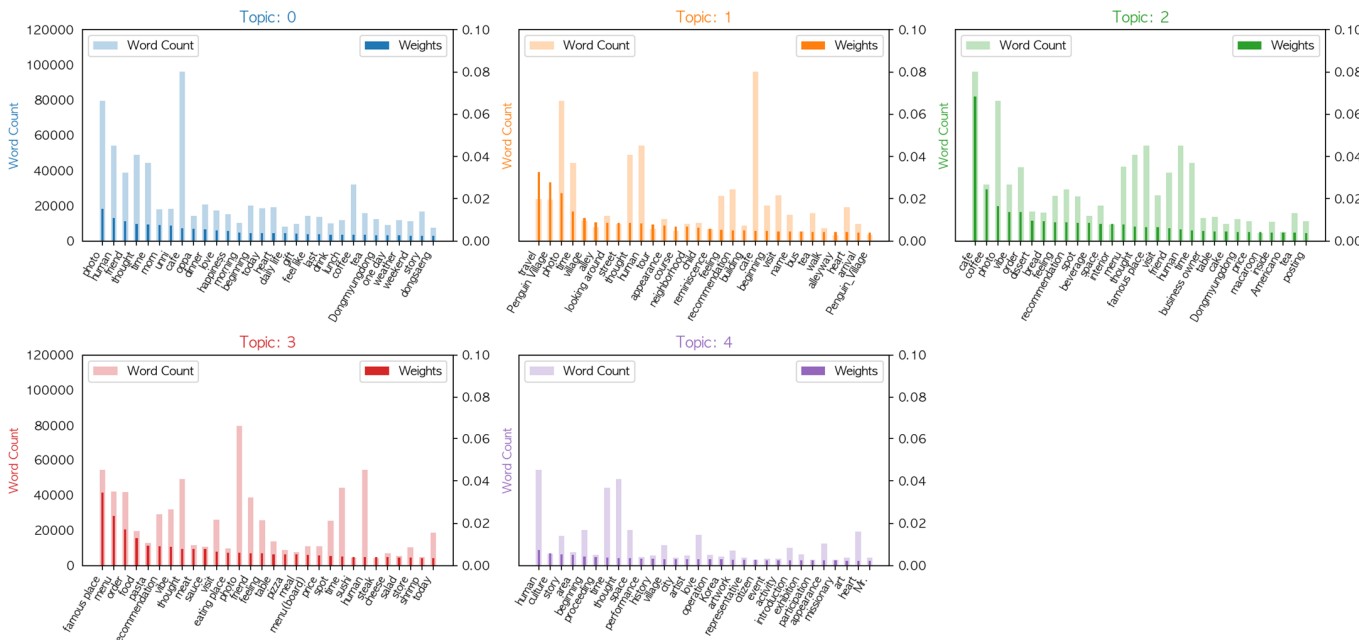

**Figure 3.** Importance (weight) of keywords by each topic and total frequency of keywords for all data.

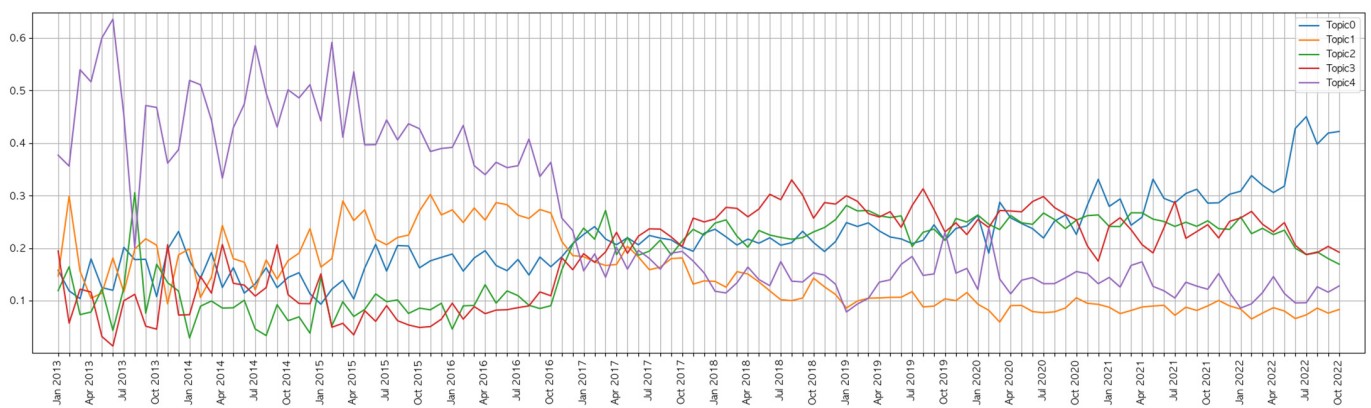

**Figure 4.** Trends for each topic from 2013 to 2022.

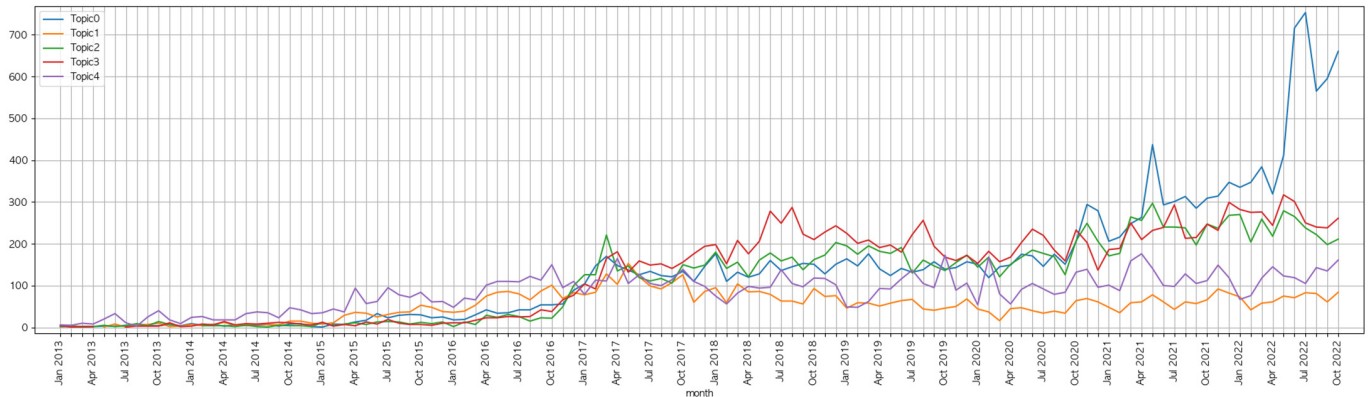

**Figure 5.** The count of blog posting by topic and month.

*4.2. Topic Changes by Period*

4.2.1. Stage 1: Flourishment (January 2013–October 2016)

The first phase of urban regeneration was from January 2013 to October 2016. In this stage, the local government worked to revitalize the neighborhood by commodifying its historical and cultural resources, and the area was named Yanglim History and Culture Village. In tandem with these initiatives, the local residents revamped the streets and alleys in the neighborhood through voluntary and cooperative participation. Later on, the Yanglim neighborhood was named Penguin Village and gained fame, being reported on in mass media and going viral.

First of all, the authors identified the theme of Stage 1 as cultural heritage and tourism. In this stage, the keywords related to this theme are history, art, and travel, rather than cafés and hot spots. Specifically, the most salient keywords determined through LDA modeling included *photo* (4927 times), *human* (4673), *travel* (3304), *thought* (3179), *culture* (2572), *Penguin Village* (2550), *café* (2289), *village* (2150), *story* (2049), and *area* (1833). Other terms are as follows: *space, friend, artwork, alley, proceeding, looking around, Korea, coffee, art, Mr., experience, missionary, church, artist, order, menu, movie, China, (musical) performance, and music* (Table 1).

In addition, ten topics were identified by perplexity and coherence scores in this stage, bringing up more detailed keywords. The authors organized these ten topics into four themes: (1) culture, travel, and area (36.3%); (2) historic and cultural heritage and the evolution of Penguin Village (23.8%); (3) the creation of the neighborhood (20.9%); and (4) cafés and daily life (19.0%). These classifications are illustrated in Figure 6 and Appendix B.

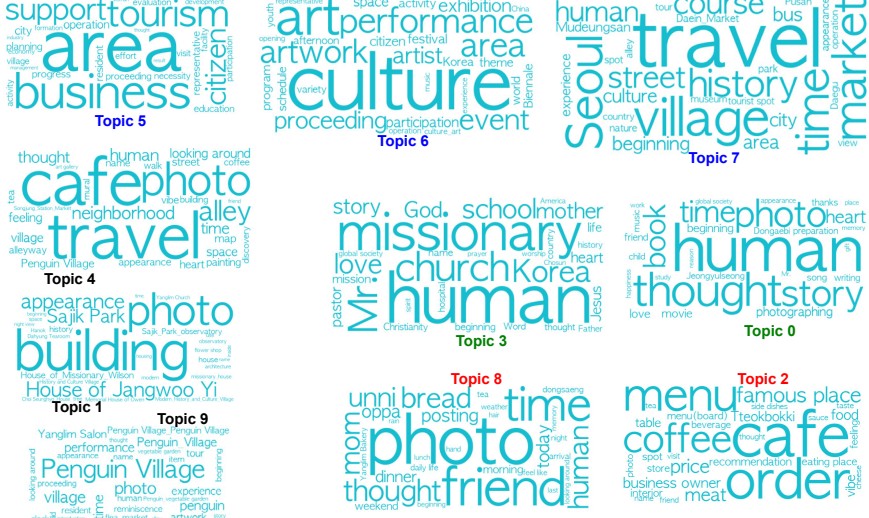

**Figure 6.** Word cloud for Stage 1. Topic numbers under the same theme have the same color for readability.

The first theme (culture, travel, and area) is related to Topic 5 (8.3%), Topic 6 (12.1%), and Topic 7 (15.9%). The relevant keywords with the highest importance scores are (1) *area* (0.0125), *business* (0.0123), *tourism* (0.0087), *support* (0.0086), and *citizen* (0.0074) in Topic 5; (2) *culture* (0.0286), *art* (0.0109), *performance* (0.0107), *area* (0.0101), and *artwork* (0.0092) in Topic 6; and (3) *travel* (0.0230), *village* (0.0077), *Seoul* (0.0063), *market* (0.0059), and *time* (0.0059) in Topic 7.

The second theme, historic and cultural heritage and the evolution of Penguin Village, is associated with three topics (i.e., Topic 1, 4, and 9). The emerging keywords with the highest importance are as follows: (1) *building* (0.0268), *photo* (0.0148), *House of Jangwoo Yi* (0.0116), *appearance* (0.0115), and *Sajik Park* (0.0099) in Topic 1 (7.7%); (2) *café* (0.0207), *travel* (0.0199), *photo* (0.0169), *alley* (0.0144), and *neighborhood* (0.0143) in Topic 4 (9.5%); and

(3) *Penguin Village* (0.0595), *photo* (0.0213), *village* (0.0185), *time* (0.0136), and *penguin* (0.0128) in Topic 9 (6.6%).

The third theme of the Yanglim neighborhood is the creation of the neighborhood story, with two topics (Topic 0 and 3). In addition to commodifying the heritage of the neighborhood, the enthusiastic storytelling about the neighborhood attracted unique small businesses. The relevant keywords by the highest importance scores are *human* (0.0205), *thought* (0.0181), *photo* (0.0179), *story* (0.0178), and *time* (0.0111), as well as *book*, *heart*, *photographing*, *love*, and *movie* (Topic 0; 9.2%). In Topic 3 (11.7%), *human* (0.0229), *missionary* (0.0120), *church* (0.0116), *Mr.* (0.0112), and *Korea* (0.0095) are relevant terms.

The last theme is about cafés and daily life. This topic accounts for merely 19% of the data (9.1% for Topic 2 and 9.9% for Topic). The important keywords are (1) *café* (0.0198), *order* (0.0150), *menu* (0.0134), *coffee* (0.0129), and *famous place* (0.0129; Topic 2) and (2) *photo* (0.0235), *friend* (0.0128), *time* (0.0113), *thought* (0.0107), and *bread* (0.0098; Topic 9). On the whole, the most important keyword in Topic 9 is *Penguin Village* (0.0595), as illustrated in Figure 7.

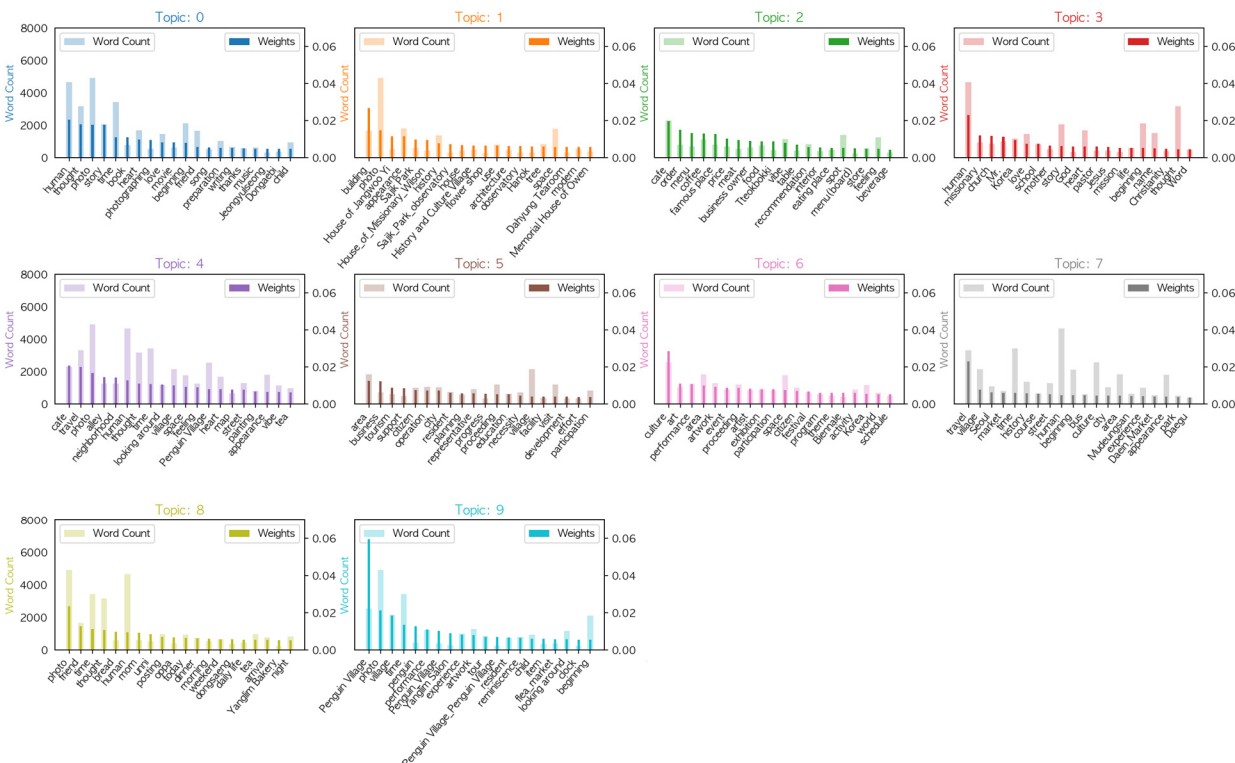

**Figure 7.** Importance (weight) of keywords by each topic and total frequency of the keywords for Stage 1.

4.2.2. Stage 2: Maturation (November 2016–February 2020)

The strategy for urban regeneration utilizing cultural and historical resources in the neighborhood became successful and attracted both locals and tourists. The newly opened local businesses with their own distinctiveness, such as restaurants and cafés, led the change in the neighborhood identity. Thus, the salient keywords in this stage are related to cafés and famous eateries and to their own vibes. The top keywords categorized by saliency are as follows: *café* (37,563), *photo* (31,725), *famous place* (20,894), *menu* (18,836), *order* (18,623), *friend* (14,871), *vibe* (14,224), *coffee* (12,204), *recommendation* (11,334), and *travel* (11,028). *Feeling, visit, food, bread, table, dessert, pasta, village, beverage, meat, menu (board), oppa*, etc., are included as well (Table 1). The theme for Stage 2 was hot spots and cultural tourism.

In this stage, the authors identified five different topics by perplexity and coherence scores; these are grouped into three unique themes (Figure 8 and Appendix C). The first

theme is daily life (Topic 0; 21.8%). The second theme is about cultural activities and tourism. With an unreproducible identity as a cultural and historical place, the summed ratio of Topic 2 (Penguin Village and travel) and Topic 1 (culture and relevant activities) entries occupies one third (31.2%) of the total. The third theme is an identity as a hot spot with good eateries and cafés (Topic 3 and 4; 47.0%). The topics for famous eateries (Topic 3; 24.5%) and cafés (Topic 4; 22.5%) are the most relevant topics, sharing fifteen keywords (e.g., *menu*, *photo*, and *vibe*) among the top thirty and occupying almost half of the topics (47%).

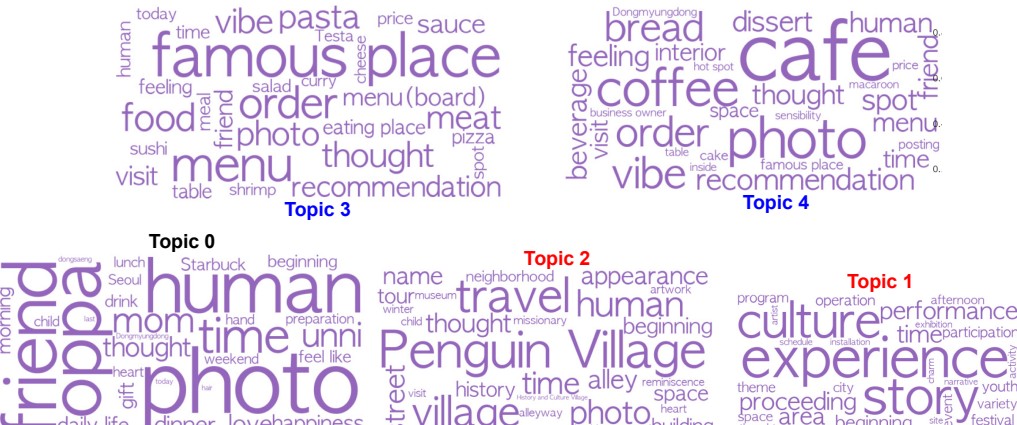

**Figure 8.** Word clouds for Stage 2. Topic numbers under the same theme have the same color for readability.

First of all, daily-life-related themes or topics account for 21.8% of the data; users' loved ones, daily activities, and feelings are all mentioned (i.e., *photo, human, friend, oppa, time, mom, unni, thought, love, dinner, happiness, daily life, gift, morning, beginning, Starbucks, feel like, drink, preparation, weekend, Seoul, lunch, heart, hand, child, Dongmyungdong, last, hair, dongsaeng* (younger people of similar age or in a blood relationship), and *today*).

Second, under the theme of cultural activities and tourism, Topic 2 (Penguin Village and travel; 18.4%) illustrates physical characteristics of the neighborhood, with keywords including *Penguin Village* (0.0190), *travel* (0.0168), *village* (0.0129), *human* (0.0099), *photo* (0.0079), *time* (0.0069), *street* (0.0057), *thought* (0.0056), *alley* (0.0056), and *appearance* (0.0049). Other keywords include *name, beginning, tour, space, history, building, neighborhood, looking around, artwork, heart, winter, child, missionary, tourist spot, alleyway, course, visit, reminiscence, museum,* and *History and Culture Village*. The keywords (Topic 1; 12.8%) indicating culture and relevant activities contain *experience* (0.0084), *story* (0.0069), *culture* (0.0068), *performance* (0.0061), *proceeding* (0.0061), *time* (0.0058), *area* (0.0056), *beginning* (0.0048), *participation* (0.0047), and *youth* (0.0047). Other culture-related keywords are *operation, festival, theme, program, event, variety, space, city, afternoon, thought, activity, representative, preparation, charm, site, artist, exhibition, installation, schedule,* and *narrative*.

Last, under the theme of a hot spot with good eateries and cafés, *menu* and *types of foods* are important keywords, reflecting the popularity of specific restaurants in this area in Topic 3 (famous eateries). The most important related terms are *famous place* (0.0284), *menu* (0.0189), *order* (0.0146), *food* (0.0104), *thought* (0.0083), *photo* (0.0083), *pasta* (0.0080), *vibe* (0.0079), *meat* (0.0078), and *recommendation* (0.0072). Other relevant keywords are as follows: *sauce, visit, menu (board), friend, eating place, feeling, table, pizza, human, time, meal, sushi, spot, salad, price, today, shrimp, cheese, curry,* and *Testa* (a restaurant in Yanglim). The keywords with the highest importance scores for famous cafés (Topic 4) include *café* (0.0558), *photo* (0.0217), *coffee* (0.0177), *vibe* (0.0119), *order* (0.0109), *bread* (0.0103), *recommendation* (0.0076), *spot* (0.0073), *feeling* (0.0073), and *dessert* (0.0069). The other keywords were listed in the following order: *thought, human, beverage, friend, menu, time, visit, interior, space, famous place, cake, posting, price, Dongmyungdong, hot spot, business owner, table, inside, macaroon,* and *sensibility*. Dongmyungdong, another revitalized neighborhood with

its own success, is located near Yanglim, and the two places were frequently mentioned or visited together.

When looking at the importance of keywords from Stage 2 (i.e., maturation), *café* is the highest, followed by *famous place*, *photo*, and *Penguin Village* from Stage 2 (Figure 9). During this stage, the Yanglim neighborhood became famous for its unique cafés and restaurants, appealing to a growing number of tourists and locals. Gentrification was highlighted in the mass media as evidence of this fame.

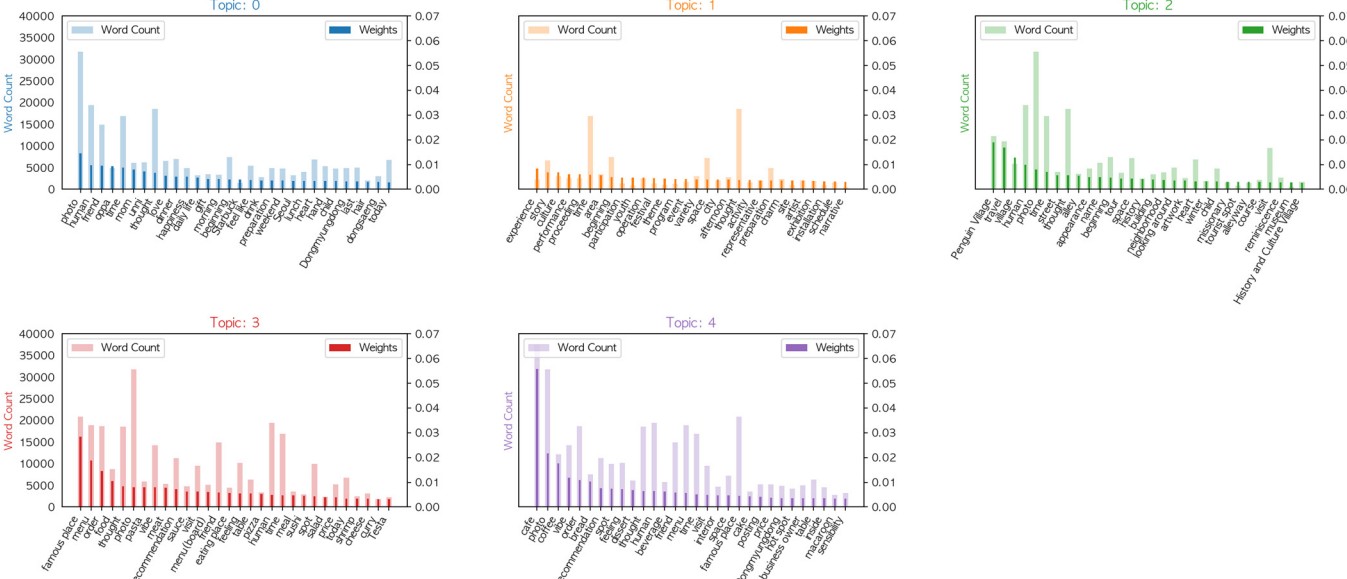

**Figure 9.** Importance (weight) of keywords by each topic and total frequency of the keywords for Stage 2.

### 4.2.3. Stage 3: COVID-19 (March 2020–October 2022)

The third phase of urban regeneration was during the COVID-19 pandemic. The first confirmed case was in January 2020; there were nine confirmed cases in Gwangju by February 2020 [51]. The initiatives and activities of the governments and local businesses related to tourism slowed down to suppress the spread of COVID-19. Several of local entrepreneurs closed down their own businesses due to the pressure of rent increases, very slow business, and a lack of capital. In accordance with these changing environments, fifteen salient keywords in this phase were associated with cafés, famous places, and daily life rather than travel and activities of local artists. The keywords include *café* (53,767), *famous place* (29,929), *order* (22,162), *menu* (21,439), *coffee* (17,525), *recommendation* (16,528), *vibe* (15,644), *visit* (15,193), *feeling* (14,186), *spot* (13,719), *space* (11,321), *unni* (11,066), *mom* (10,753), *dessert* (10,391), and *food* (10,111; Table 1). Thus, the theme was identified as daily life and a hot spot with loved ones.

Through LDA modeling, six topics were identified, with three major themes with two individual topics each, after computing perplexity and coherence scores. The three main themes were daily life (52.3%), cafés and local hot spots (34.0%), and travel, art, and local stories (14.7%; Figure 10 and Appendix D).

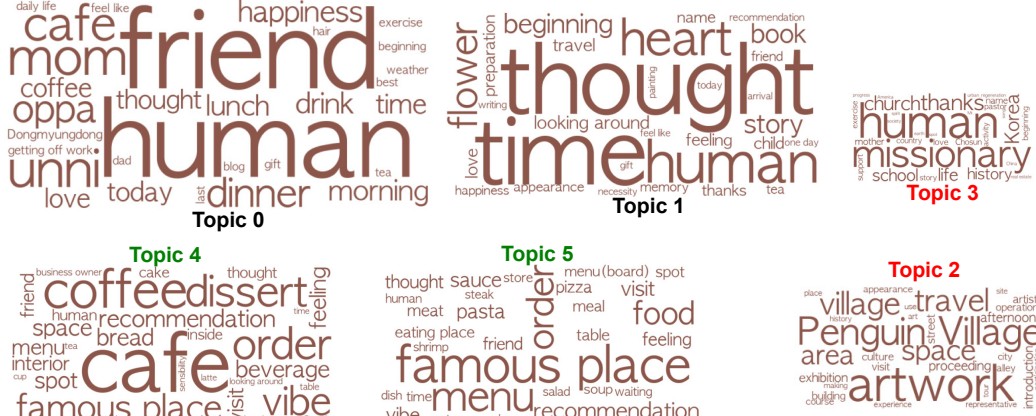

**Figure 10.** Word clouds for Stage 3. Topic numbers under the same theme have the same color for readability.

First of all, for the main theme, daily life, stories about loved ones in the neighborhood (Topic 0; 29.0%) were the most common postings during this phase. The most important ten keywords were *friend* (0.0102), *human* (0.0098), *mom* (0.0094), *unni* (0.0094), *café* (0.0080), *dinner* (0.0078), *oppa* (0.0075), *happiness* (0.0051), *morning* (0.0049), and *drink* (0.0049; Figure 11). Along with loved ones and activities, other keywords expressing feelings and happiness (e.g., *lunch, time, today, thought, coffee, love, Dongmyungdong, exercise, gift, getting off work, feel like, dad, dog, blog, last, best, beginning, daily life, hair, weather,* and *tea*) were salient. For Topic 1 (activities in the neighborhood; 22.3%), keywords related to neighborhood businesses and activities, including *book, flower, writing, tea,* and *painting,* were salient. The top ten keywords were *thought* (0.0164), *time* (0.0164), *human* (0.0164), *heart* (0.0164), *flower* (0.0164), *beginning* (0.0164), *story* (0.0164), *book* (0.0164), *looking around* (0.0164), and *travel* (0.0164). Other relevant terms were *love, feeling, child, preparation, name, thanks, appearance, friend, tea, memory, happiness, recommendation, writing, necessity, arrival, painting, gift, feel like,* and *one day.*

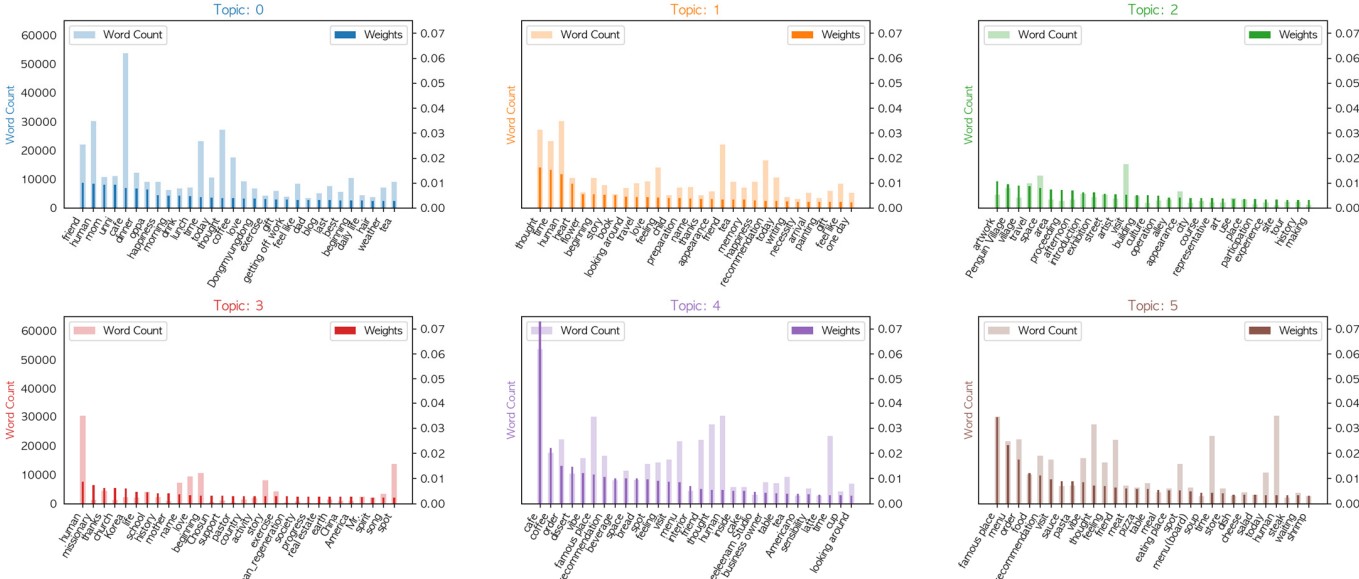

**Figure 11.** Importance (weight) of keywords by each topic and total frequency of the keywords for Stage 3.

For the second theme, keywords associated with cafés and local hot spots are illustrated. For Topic 5 (famous places and food; 17.2%), western food such as pizza and pasta

along with famous places and the vibes were all mentioned. It seems that this neighborhood is famous for western food rather than local food; as western food is more expensive than local food, this is highly likely to be evidence of gentrification in Yanglim. The ten most important keywords were *famous place* (0.0343), *menu* (0.0233), *order* (0.0176), *food* (0.0122), *recommendation* (0.0112), *visit* (0.0096), *sauce* (0.0090), *pasta* (0.0089), *vibe* (0.0085), and *thought* (0.0071). There were twenty additional salient keywords such as *feeling*, *friend*, *meat*, *pizza*, *table*, *meal*, *eating place*, *spot*, *menu (board)*, *soup*, *time*, *store*, *dish*, *cheese*, *salad*, *today*, *human*, *steak*, *waiting*, and *shrimp*. In topic 4 (16.8%), keywords about cafés and the vibe emerged. The keywords included *café* (0.0730), *coffee* (0.0220), *order* (0.0150), *dessert* (0.0146), *vibe* (0.0122), *famous place* (0.0115), *recommendation* (0.0106), *beverage* (0.0101), *space* (0.0100), and *bread* (0.0100). There are twenty keywords about *feelings*, *friends*, and *thoughts* as well. The Yanglim neighborhood has worked to combine art and local business. As a successful example, *Lee, Leenam studio* appeared, which is a café-cum-media art studio exhibiting the artwork of the media artist *Lee, Leenam*.

Travel, art, and local stories are the last theme. Topic 2—travel, Penguin Village, and art—makes up 9.5% of the keywords. Although the influx of tourist traffic slowed down in this phase, artwork, Penguin Village, and travel remained the top ranked for this topic. The importance of each keyword was as follows: *artwork* (0.0107), *Penguin Village* (0.0096), *village* (0.0091), *travel* (0.0089), *space* (0.0080), *area* (0.0074), *proceeding* (0.0073), *afternoon* (0.0073), *introduction* (0.0071), and *exhibition* (0.0063). In addition to art and culture, space-related keywords included *street*, *artist*, *visit*, *building*, *culture*, *operation*, *alley*, *appearance*, *city*, *course*, *representative*, *art*, *use*, *place*, *participation*, *experience*, *site*, *tour*, *history*, and *making*. Topic 3 was about the local story and urban regeneration (5.2%). As a heritage site of Christianity, relevant terms were *human* (0.0087), *missionary* (0.0074), *thanks* (0.0063), *church* (0.0062), *Korea* (0.0061), *life* (0.0047), *school* (0.0044), *history* (0.0042), *mother* (0.0041), and *name* (0.0038). Keywords such as *love, beginning, Chosun* (the last imperial dynasty of Korea), *support, pastor, country, activity, story, exercise, urban regeneration, society, progress, real estate, earth, China, America, Mr., spirit, song*, and *spot* were included as well.

## 5. Discussion

Analyzing big data in urban regeneration areas helps policy makers to better understand citizens' thoughts and needs and how they position themselves within the cities and neighborhoods. Comprehending neighborhood identity and its changing nature offers a framework for this analysis. Local government policies making good use of historical and cultural resources and citizen participation have become drivers of neighborhood changes and have gradually transformed neighborhood identities, as has the COVID-19 pandemic more recently. In this study, the authors have examined how the identity of the Yanglim area of Gwangju, Korea has changed through urban regeneration initiatives and citizens' participation and interactions. The authors divided this neighborhood identity into three phases using trend analysis: (1) January 2013 to October 2016 (stage 1: flourishment), (2) November 2016 to February 2020 (stage 2: maturation), and (3) March 2020 to October 2022 (stage 3: COVID-19).

The topics included the issues around events, businesses, emotions and sentiments, users, activities, history, art, and culture in the neighborhood [52]. The topics and salient keywords identified have changed over time.

In the beginning, Yanglim was unnoticed by locals and tourists. However, cooperation between the local government and residents brought about successful urban regeneration. The local government steered the maintenance and improvement of the neighborhood environment and made use of storytelling through historical and cultural resources (i.e., culture-led urban regeneration). In addition, the voluntary involvement of the local residents and the cooperation of local artists in beautifying a burnt house site and the surrounding neighborhood helped it acquire its current reputation as Penguin Village, the new name of the Yanglim area. This cooperation between government and local residents

played a crucial role in burgeoning urban regeneration and a new identity introduced by cultural and historical resources for this area.

Second, the growing demands from new clientele became an important driver of neighborhood change [53]. Thanks to the influx of tourists and local visitors for sightseeing, small business entrepreneurs opened shops with their own featured items and foods. During the first phase, businesses taking advantage of affordable rents (e.g., photo studios, flower shops, wedding shops, and guest houses as well as coffee shops, bakeries, and restaurants) were the primary businesses in the area. For example, wedding shops, guest houses, and photo studios lured local customers who desired special experiences combining cultural and modern heritage. As time went by, newly opened food-related businesses such as coffee shops, bakeries, and restaurants became the predominant businesses due to a demand for these businesses by a new population in the area, such as tourists and young locals. As a result, the neighborhood's identity changed to a hot spot going through commercialization, with a great many top-rated and must-see restaurants and cafés with their own unique tastes and moods. The distinctive characteristic of urban regeneration projects in Korea is commercialization by converting residential buildings to commercial buildings such as cafés and restaurants. This conversion changes neighborhoods into tourist spots, though this brings up gentrification issues [54]. These food-related businesses function as a proxy of local economic change associated with gentrification in neighborhoods, providing real-time and up-to-date insights for gentrification before official statistics are released [55]. As expected, this identity change in Yanglim has led to gentrification issues, with the growing increase of rents reported in the news media. Thus, the local government, landlords, and small entrepreneurs (i.e., tenants) made an agreement to prevent a dramatic increase in rent in 2019, promising a government subsidy [56]. Government statistics released later showed that the number of food-related businesses that closed down increased by 50% from 2018 to 2019, confirming these news reports (Table 2). On top of these endeavors, the local government has worked to attract local artists by designating a block as the Yanglim Culture Park (4071 m$^2$, 1077 m$^2$ for buildings) created between 2018 and 2020 [57]. All of the buildings built or renovated with the Hanok architecture type are occupied by fourteen different studios for arts and crafts [58].

**Table 2.** The number of small businesses that closed down between 2013 and 2022 in Yanglim.

|  | 2013 | 2014 | 2015 | 2016 | 2017 | 2018 | 2019 | 2020 | 2021 | 2022 |
|---|---|---|---|---|---|---|---|---|---|---|
| N of Businesses closed down (café, bakery, restaurant etc.) | 11 | 8 | 8 | 7 | 11 | 14 | 21 | 23 | 20 | 27 |

Data obtained from https://www.localdata.go.kr/main.do (accessed on 22 March 2023).

Lastly, in addition to the gentrification issue, COVID-19 has propelled unexpected neighborhood changes. Yanglim's identity as a hot spot for tourists and locals has been weakened due to the worldwide pandemic. Small business owners (tenants) with insufficient capital closed down their businesses due to slow business as well as rising rents and real estate prices. As seen in the data in Table 2, the number of businesses shutting down has gradually grown since 2017, and almost doubled in 2022 compared to 2018 (i.e., from 14 to 27). As the Yanglim neighborhood has become less famous as a hot spot for the young and tourists following the COVID-19 pandemic, the changing nature of the neighborhood is identified by the majority of blog posts as a place for individuals and their loved ones. Following the COVID-19 pandemic, this neighborhood now lies between the point of retrieving its previous reputation as a renowned tourist spot with an abundant heritage and art studios and a hot spot with rich cafés and eateries drawing the attention of the young. It is likely that gentrification is one of the barriers to winning back its fame. Because the Yanglim Culture Park opened during the COVID-19 outbreak, local artists have stated that initiatives embracing their activities (e.g., one-day classes or festivals) are necessary to ap-

peal to more potential tourists and locals. However, the Korean central government has cut the budget for urban regeneration projects, and currently the local government feels pressure to create self-sustainable neighborhoods [59]. Subsequently, it is important to strategize for economically and socially sustainable neighborhoods in the era of depopulation and degentrification.

## 6. Conclusions and Limitations

This study has attempted to analyze how neighborhood identity has changed through linguistic patterns using computational techniques (i.e., text mining). User-generated data provide insights about urban regeneration. Such insights can be valuable for improving current situations and planning for economically and socially sustainable neighborhoods and cities [8].

On the one hand, urban regeneration in the Yanglim area was considered to be successful overall. In the first phase, the identity of the Yanglim neighborhood in Gwangju, Korea, was formed by different stakeholders. The activities of policymakers and local residents led to a shared identity known as the Yanglim History and Culture Village, or Penguin Village. Starting with an identity as a tourist spot, the identity of this area evolved into a hot spot with trendy and unique cafés and restaurants, drawing attention from a considerable number of tourists and local young adults. On the other hand, gentrification became an issue, similar to other urban regeneration sites. The outbreak of the COVID-19 pandemic and the resulting decrease in visitors and tourists changed the neighborhood's identity to a place for everyday life. The dominant blog postings during that time were about time and activities spent with loved ones such as friends and family members in the neighborhood. This means that the neighborhood has been experiencing degentrification, losing its reputation as a hot spot due to economic crisis [54]. Consequently, it appears that the process of enabling this neighborhood to be economically and socially sustainable has been challenged by successive experiences of gentrification and degentrification.

While blog data allow us to listen to citizens' voices through massive volumes of data, it has possible biases. First of all, blog postings are more likely to be used to promote businesses due to entrepreneurs or compensated service users uploading topics which generate a high number of keywords rather than personal online postings. Second, analyzing blog data may not adequately capture the viewpoints of residents and small business owners in the area, especially those who experience displacement and gentrification. Whereas displacement and gentrification issues may show up in micro-level data analysis, LDA modeling, as a technique for macro-level analysis, may not be able to capture these issues. In addition, the majority of blog or social media users tend to emphasize the positive side of their lives [52]. Thus, considering other diverse research methods is recommended for further research.

**Author Contributions:** H.Y.Y. contributed to the entire research process, including conceptualization, data collection, writing, and revision. H.-a.K. contributed to funding acquisition, project administration, supervision, and review of the paper. All authors have read and agreed to the published version of the manuscript.

**Funding:** This research was funded by the Ministry of Education of the Republic of Korea through the National Research Foundation of Korea (NRF-2019R1I1A3A01061072).

**Institutional Review Board Statement:** Not applicable.

**Informed Consent Statement:** Not applicable.

**Data Availability Statement:** Data is not available for confidentiality.

**Acknowledgments:** The authors acknowledge financial support from the funder and comments from three unknown reviewers for the improvement of this paper.

**Conflicts of Interest:** The authors declare no conflict of interest.

**Appendix A. Most Important Terms and Blog Posting Numbers by Topic and Year**

| T | Keyword1 | Keyword2 | Keyword3 | Keyword4 | Keyword5 | 2013 | 2014 | 2015 | 2016 | 2017 | 2018 | 2019 | 2020 | 2021 | 2022 | Sum | % of tokens |
|---|---|---|---|---|---|---|---|---|---|---|---|---|---|---|---|---|---|
| 0 | photo | human | friend | thought | time | 42 | 69 | 240 | 515 | 1601 | 1690 | 1758 | 2160 | 3532 | 5085 | 16,692 | 33.2 |
| 1 | travel | Penguin Village | photo | time | village | 47 | 103 | 386 | 859 | 1237 | 933 | 652 | 530 | 743 | 689 | 6179 | 10.3 |
| 2 | café | coffee | photo | vibe | order | 56 | 37 | 124 | 341 | 1649 | 1940 | 2017 | 2071 | 2832 | 2364 | 13,431 | 17.5 |
| 3 | famous place | menu | order | food | pasta | 32 | 101 | 105 | 385 | 1801 | 2658 | 2375 | 2235 | 2805 | 2684 | 15,181 | 18.6 |
| 4 | human | culture | story | area | beginning | 188 | 354 | 781 | 1204 | 1365 | 1178 | 1165 | 1176 | 1476 | 1192 | 10,079 | 20.4 |

**Appendix B. Most Important Terms by Topic for All Data**

| Topic 0: Daily Life (33.2%) | | Topic 1: Tour and Culture (20.4%) | | Topic 2: Café and Hot Spot (18.6%) | | Topic 3: Food and Hot Spot (17.5%) | | Topic 4: Culture and Citizen Participation (10.3%) | |
|---|---|---|---|---|---|---|---|---|---|
| Keywords | Importance | Keywords | Importance | Keywords | Importance | Keywords | Importance | Keywords | Importance |
| photo | 0.0153 | travel | 0.0326 | café | 0.0684 | famous place | 0.0344 | human | 0.0073 |
| human | 0.0109 | Penguin Village | 0.0278 | coffee | 0.0244 | menu | 0.0234 | culture | 0.0058 |
| friend | 0.0095 | photo | 0.0225 | photo | 0.0165 | order | 0.0170 | story | 0.0054 |
| thought | 0.0080 | time | 0.0139 | vibe | 0.0137 | food | 0.0131 | area | 0.0052 |
| time | 0.0077 | village | 0.0109 | order | 0.0137 | pasta | 0.0093 | beginning | 0.0043 |
| mom | 0.0075 | alley | 0.0088 | dessert | 0.0096 | recommendation | 0.0091 | proceeding | 0.0041 |
| unni | 0.0073 | looking around | 0.0087 | bread | 0.0092 | vibe | 0.0090 | time | 0.0037 |
| café | 0.0060 | street | 0.0086 | feeling | 0.0089 | thought | 0.0080 | thought | 0.0035 |
| oppa | 0.0057 | thought | 0.0085 | recommendation | 0.0087 | meat | 0.0078 | space | 0.0034 |
| dinner | 0.0056 | human | 0.0082 | spot | 0.0085 | sauce | 0.0078 | performance | 0.0034 |
| love | 0.0050 | tour | 0.0079 | beverage | 0.0085 | visit | 0.0067 | history | 0.0033 |
| happiness | 0.0047 | appearance | 0.0073 | space | 0.0082 | eating place | 0.0061 | village | 0.0031 |
| morning | 0.0041 | course | 0.0067 | interior | 0.0081 | photo | 0.0061 | city | 0.0031 |
| beginning | 0.0037 | neighborhood | 0.0067 | menu | 0.0079 | friend | 0.0060 | artist | 0.0030 |
| today | 0.0037 | child | 0.0062 | thought | 0.0067 | feeling | 0.0060 | love | 0.0030 |
| heart | 0.0036 | reminiscence | 0.0059 | famous place | 0.0066 | table | 0.0055 | operation | 0.0030 |
| daily life | 0.0036 | feeling | 0.0053 | visit | 0.0065 | pizza | 0.0054 | Korea | 0.0030 |

| Topic 0: Daily Life (33.2%) | | Topic 1: Tour and Culture (20.4%) | | Topic 2: Café and Hot Spot (18.6%) | | Topic 3: Food and Hot Spot (17.5%) | | Topic 4: Culture and Citizen Participation (10.3%) | |
|---|---|---|---|---|---|---|---|---|---|
| Keywords | Importance | Keywords | Importance | Keywords | Importance | Keywords | Importance | Keywords | Importance |
| gift | 0.0036 | recommendation | 0.0050 | friend | 0.0060 | meal | 0.0054 | artwork | 0.0028 |
| feel like | 0.0033 | building | 0.0050 | human | 0.0056 | menu(board) | 0.0052 | representative | 0.0027 |
| last | 0.0032 | café | 0.0047 | time | 0.0050 | price | 0.0048 | citizen | 0.0027 |
| drink | 0.0030 | beginning | 0.0046 | business owner | 0.0049 | spot | 0.0047 | event | 0.0026 |
| lunch | 0.0030 | visit | 0.0046 | table | 0.0046 | time | 0.0043 | activity | 0.0026 |
| coffee | 0.0029 | name | 0.0046 | cake | 0.0044 | sushi | 0.0041 | introduction | 0.0025 |
| tea | 0.0029 | bus | 0.0045 | Dongmyungdong | 0.0043 | human | 0.0041 | exhibition | 0.0025 |
| Dongmyungdong | 0.0028 | tea | 0.0043 | price | 0.0042 | steak | 0.0041 | participation | 0.0025 |
| one day | 0.0028 | walk | 0.0043 | macaroon | 0.0042 | cheese | 0.0039 | appearance | 0.0024 |
| weather | 0.0026 | alleyway | 0.0043 | inside | 0.0041 | salad | 0.0039 | missionary | 0.0024 |
| weekend | 0.0026 | heart | 0.0043 | Americano | 0.0040 | store | 0.0039 | art | 0.0024 |
| story | 0.0025 | arrival | 0.0041 | tea | 0.0039 | shrimp | 0.0036 | heart | 0.0023 |
| dongsaeng | 0.0025 | Penguin_Village | 0.0041 | posting | 0.0038 | today | 0.0035 | Mr. | 0.0022 |

## Appendix C. Most Important Terms by Topic and Theme for Stage 1

| Theme 1: Culture, Travel, and Area (36.3%) | | | | | | Theme 3: the Creation of the Neighborhood (20.9%) | | | |
|---|---|---|---|---|---|---|---|---|---|
| Topic 5 (8.3%) | | Topic 6 (12.1%) | | Topic 7 (15.9%) | | Topic 0 (9.2%) | | Topic 3 (11.7%) | |
| Keyword | Importance | Keyword | Importance | Keyword | Importance | Keyword | Importance | Keyword | Importance |
| area | 0.0125 | culture | 0.0286 | travel | 0.023 | human | 0.0205 | human | 0.0229 |
| business | 0.0123 | art | 0.0109 | village | 0.0077 | thought | 0.0181 | missionary | 0.012 |
| tourism | 0.0087 | performance | 0.0107 | Seoul | 0.0063 | photo | 0.0179 | church | 0.0116 |
| support | 0.0086 | area | 0.0101 | market | 0.0059 | story | 0.0178 | Mr. | 0.0112 |
| citizen | 0.0074 | artwork | 0.0092 | time | 0.0059 | time | 0.0111 | Korea | 0.0095 |
| operation | 0.0073 | event | 0.0088 | history | 0.0058 | book | 0.0109 | love | 0.0075 |
| city | 0.0072 | proceeding | 0.0087 | course | 0.0054 | heart | 0.0099 | school | 0.0074 |
| resident | 0.0059 | artist | 0.0083 | street | 0.0053 | photographing | 0.0094 | mother | 0.0064 |
| planning | 0.0058 | exhibition | 0.0079 | human | 0.0048 | love | 0.0083 | story | 0.0061 |
| representative | 0.0057 | participation | 0.0079 | beginning | 0.0047 | movie | 0.0082 | God | 0.0061 |
| progress | 0.0056 | space | 0.0076 | bus | 0.0047 | beginning | 0.0079 | heart | 0.006 |
| proceeding | 0.0054 | citizen | 0.0071 | culture | 0.0046 | friend | 0.0058 | pastor | 0.0059 |

| | | | | | | | | | |
|---|---|---|---|---|---|---|---|---|---|
| education | 0.0052 | festival | 0.0069 | city | 0.0045 | song | 0.0054 | Jesus | 0.0057 |
| necessity | 0.0045 | program | 0.0061 | area | 0.0045 | preparation | 0.0052 | mission | 0.0053 |
| village | 0.0041 | theme | 0.006 | Mudeungsan | 0.0043 | writing | 0.0052 | life | 0.0053 |
| facility | 0.004 | Biennale | 0.006 | experience | 0.0042 | thanks | 0.0051 | beginning | 0.0051 |
| visit | 0.004 | activity | 0.0059 | Daein_Market | 0.004 | music | 0.0048 | name | 0.0051 |
| development | 0.0039 | Korea | 0.0054 | appearance | 0.0039 | Jeongyulseong | 0.0048 | Christianity | 0.0049 |
| effort | 0.0039 | world | 0.0053 | park | 0.0038 | Dongaebi | 0.0048 | thought | 0.0046 |
| participation | 0.0038 | schedule | 0.0052 | Daegu | 0.0035 | child | 0.0047 | word | 0.0045 |
| evaluation | 0.0037 | afternoon | 0.0051 | nature | 0.0035 | appearance | 0.0047 | hospital | 0.0044 |
| activity | 0.0036 | variety | 0.0045 | tour | 0.0034 | work | 0.0042 | country | 0.004 |
| development | 0.0035 | youth | 0.0042 | spot | 0.0032 | happiness | 0.0041 | history | 0.0038 |
| economy | 0.0033 | culture_art | 0.0042 | Pusan | 0.0032 | study | 0.004 | Father | 0.0037 |
| formation | 0.0031 | operation | 0.0042 | alley | 0.0032 | reason | 0.0038 | global society | 0.0035 |
| management | 0.0031 | representative | 0.0041 | view | 0.003 | gift | 0.0036 | worship | 0.0033 |
| thought | 0.0031 | experience | 0.004 | tourist spot | 0.003 | place | 0.0035 | spirit | 0.0033 |
| result | 0.0031 | opening | 0.004 | museum | 0.003 | global society | 0.0034 | America | 0.0033 |
| woman | 0.003 | music | 0.0039 | country | 0.0029 | memory | 0.0034 | Chosun | 0.003 |
| industry | 0.003 | China | 0.0038 | operation | 0.0028 | Mr. | 0.0033 | prayer | 0.0029 |

| Theme 2: historic and cultural heritages and evolution of Penguin Village (23.8%) | | | | | | Theme 4: cafés and daily life (19.0%) | | | |
|---|---|---|---|---|---|---|---|---|---|
| Topic 1 (7.7%) | | Topic 4 (9.5%) | | Topic 9 (6.6%) | | Topic 2 (9.1%) | | Topic 8 (9.9%) | |
| building | 0.0268 | café | 0.0207 | Penguin Village | 0.0595 | café | 0.0198 | photo | 0.0235 |
| photo | 0.0148 | travel | 0.0199 | photo | 0.0213 | order | 0.015 | friend | 0.0128 |
| House of Jangwoo Yi | 0.0116 | photo | 0.0169 | village | 0.0185 | menu | 0.0134 | time | 0.0113 |
| appearance | 0.0115 | alley | 0.0144 | time | 0.0136 | coffee | 0.0129 | thought | 0.0107 |
| Sajik Park | 0.0099 | neighborhood | 0.0143 | penguin | 0.0128 | famous place | 0.0129 | bread | 0.0098 |
| House_of_Missioary_Wilson | 0.0096 | human | 0.0127 | performance | 0.0111 | price | 0.0102 | human | 0.0095 |
| history | 0.0077 | thought | 0.0111 | Penguin_Village | 0.0102 | meat | 0.0094 | mom | 0.0094 |
| Sajik_Park_observatory | 0.0073 | time | 0.0108 | Yanglim Salon | 0.0089 | business owner | 0.009 | unni | 0.0086 |
| house | 0.0068 | looking around | 0.0105 | experience | 0.0082 | food | 0.0088 | posting | 0.0069 |
| History and Culture Village | 0.0066 | village | 0.0099 | artwork | 0.0081 | Tteokbokki | 0.0087 | oppa | 0.0068 |
| flower shop | 0.0066 | space | 0.0093 | tour | 0.0071 | vibe | 0.008 | today | 0.0065 |

| | | | | | | | | | |
|---|---|---|---|---|---|---|---|---|---|
| use | 0.0065 | feeling | 0.009 | Penguin Village_Penguin Village | 0.007 | table | 0.0071 | dinner | 0.0064 |
| architecture | 0.0063 | Penguin Village | 0.0081 | resident | 0.0067 | recommendation | 0.0059 | morning | 0.006 |
| observatory | 0.0062 | heart | 0.008 | reminiscence | 0.0065 | interior | 0.0055 | weekend | 0.0059 |
| Hanok | 0.0061 | map | 0.0078 | child | 0.0061 | eating place | 0.0052 | dongsaeng | 0.0058 |
| tree | 0.0061 | street | 0.0077 | item | 0.0059 | spot | 0.0052 | daily life | 0.0055 |
| space | 0.0059 | painting | 0.0069 | flea_market | 0.0059 | menu (board) | 0.005 | tea | 0.0054 |
| Dahyung Tearoom | 0.0057 | appearance | 0.0066 | looking around | 0.0058 | store | 0.0048 | arrival | 0.0054 |
| modern | 0.0057 | vibe | 0.0065 | clock | 0.0056 | feeling | 0.0048 | Yanglim Bakery | 0.0052 |
| Memorial House of Owen | 0.0057 | tea | 0.0063 | beginning | 0.0056 | beverage | 0.0043 | night | 0.0052 |
| missionary_house | 0.0056 | name | 0.0062 | human | 0.0056 | photo | 0.004 | feel like | 0.0049 |
| Modern_History_and_Culture_Village | 0.0052 | alleyway | 0.0061 | proceeding | 0.0055 | thought | 0.0039 | weather | 0.0048 |
| inside | 0.0052 | coffee | 0.0053 | appearance | 0.0053 | friend | 0.0038 | hand | 0.0044 |
| Choi Se-unghyo_House | 0.0051 | walk | 0.0052 | name | 0.0051 | tea | 0.0037 | rain | 0.0041 |
| Yanglim Church | 0.005 | building | 0.0049 | Penguin_vegetable garden | 0.0051 | side dishes | 0.0037 | hair | 0.004 |
| time | 0.0046 | discovery | 0.0048 | alley | 0.005 | name | 0.0035 | beginning | 0.0039 |
| beginning | 0.0045 | mural | 0.0046 | vegetable garden | 0.0049 | visit | 0.0034 | lunch | 0.0037 |
| night view | 0.0044 | Songjung_Station_Market | 0.0045 | story | 0.0049 | cheese | 0.0033 | last | 0.0037 |
| name | 0.0044 | friend | 0.0044 | introduction | 0.0048 | taste | 0.0033 | looking around | 0.0036 |
| housing | 0.0043 | art gallery | 0.0044 | thought | 0.0048 | sauce | 0.0033 | memory | 0.0036 |

**Appendix D. Most Important Terms by Topic and Theme for Stage 2**

| Theme 1: Daily Life (21.8%) | | Theme 2: Cultural Activities and Tourism (31.2%) | | | | Theme 3: Hot Spot with Good Eateries and Cafés (47.0%) | | | |
|---|---|---|---|---|---|---|---|---|---|
| Topic 0: Daily Life (21.8%) | | Topic 1: Culture and Relevant Activities (12.8%) | | Topic 2: Penguin Village and Travel (18.4%) | | Topic 3: Famous Eateries (24.5%) | | Topic 4: Famous Cafés (22.5%) | |
| Keyword | Importance | Keyword | Importance | Keyword | Importance | Keyword | Importance | Keyword | Importance |
| photo | 0.0145 | experience | 0.0084 | Penguin Village | 0.0190 | famous place | 0.0284 | café | 0.0558 |
| human | 0.0096 | story | 0.0069 | travel | 0.0168 | menu | 0.0189 | photo | 0.0217 |
| friend | 0.0095 | culture | 0.0068 | village | 0.0129 | order | 0.0146 | coffee | 0.0177 |
| oppa | 0.0094 | performance | 0.0061 | human | 0.0099 | food | 0.0104 | vibe | 0.0119 |
| time | 0.0088 | proceeding | 0.0061 | photo | 0.0079 | thought | 0.0083 | order | 0.0109 |
| mom | 0.0081 | time | 0.0058 | time | 0.0069 | photo | 0.0081 | bread | 0.0103 |
| unni | 0.0072 | area | 0.0056 | street | 0.0057 | pasta | 0.0080 | recommendation | 0.0076 |
| thought | 0.0066 | beginning | 0.0048 | thought | 0.0056 | vibe | 0.0079 | spot | 0.0073 |
| love | 0.0056 | participation | 0.0047 | alley | 0.0056 | meat | 0.0078 | feeling | 0.0073 |
| dinner | 0.0052 | youth | 0.0047 | appearance | 0.0049 | recommendation | 0.0072 | dessert | 0.0069 |
| happiness | 0.0050 | operation | 0.0046 | name | 0.0048 | sauce | 0.0062 | thought | 0.0065 |
| daily life | 0.0048 | festival | 0.0044 | beginning | 0.0048 | visit | 0.0062 | human | 0.0065 |
| gift | 0.0042 | theme | 0.0043 | tour | 0.0044 | menu (board) | 0.0060 | beverage | 0.0062 |
| morning | 0.0041 | program | 0.0042 | space | 0.0043 | friend | 0.0059 | friend | 0.0059 |
| beginning | 0.0040 | event | 0.0042 | history | 0.0042 | eating place | 0.0057 | menu | 0.0057 |
| Starbucks | 0.0040 | variety | 0.0040 | building | 0.0040 | feeling | 0.0055 | time | 0.0050 |
| feel like | 0.0037 | space | 0.0039 | neighborhood | 0.0037 | table | 0.0055 | visit | 0.0049 |
| drink | 0.0036 | city | 0.0039 | looking around | 0.0035 | pizza | 0.0054 | interior | 0.0047 |
| preparation | 0.0035 | afternoon | 0.0039 | artwork | 0.0035 | human | 0.0050 | space | 0.0046 |
| weekend | 0.0035 | thought | 0.0037 | heart | 0.0032 | time | 0.0048 | famous place | 0.0046 |
| Seoul | 0.0034 | activity | 0.0037 | winter | 0.0031 | meal | 0.0047 | cake | 0.0043 |
| lunch | 0.0034 | representative | 0.0036 | child | 0.0031 | sushi | 0.0044 | posting | 0.0041 |

| Theme 1: Daily Life (21.8%) | | Theme 2: Cultural Activities and Tourism (31.2%) | | | | Theme 3: Hot Spot with Good Eateries and Cafés (47.0%) | | | |
| --- | --- | --- | --- | --- | --- | --- | --- | --- | --- |
| Topic 0: Daily Life (21.8%) | | Topic 1: Culture and Relevant Activities (12.8%) | | Topic 2: Penguin Village and Travel (18.4%) | | Topic 3: Famous Eateries (24.5%) | | Topic 4: Famous Cafés (22.5%) | |
| Keyword | Importance | Keyword | Importance | Keyword | Importance | Keyword | Importance | Keyword | Importance |
| heart | 0.0034 | preparation | 0.0036 | missionary | 0.0030 | spot | 0.0044 | price | 0.0037 |
| hand | 0.0033 | charm | 0.0036 | tourist spot | 0.0030 | salad | 0.0040 | Dongmyung-dong | 0.0036 |
| child | 0.0033 | site | 0.0036 | alleyway | 0.0030 | price | 0.0040 | hot spot | 0.0036 |
| Dongmyung-dong | 0.0032 | artist | 0.0035 | course | 0.0029 | today | 0.0034 | business owner | 0.0036 |
| last | 0.0032 | exhibition | 0.0034 | visit | 0.0027 | shrimp | 0.0034 | table | 0.0035 |
| hair | 0.0031 | installation | 0.0032 | reminiscence | 0.0027 | cheese | 0.0033 | inside | 0.0035 |
| dongsaeng | 0.0030 | schedule | 0.0031 | museum | 0.0027 | curry | 0.0032 | macaroon | 0.0033 |
| today | 0.0029 | narrative | 0.0030 | History and Culture Village | 0.0026 | Testa | 0.0031 | sensibility | 0.0033 |

**Appendix E. Most Important Terms by Topic and Theme for Stage 3**

| Theme 1: Daily Life (52.3%) | | | | Theme 3: Travel, Art, and Local Stories (14.7%) | | | | Theme 2: Cafés and Local Hot Spots (34.0%) | | | |
| --- | --- | --- | --- | --- | --- | --- | --- | --- | --- | --- | --- |
| Topic 0: Loved Ones in the Neighborhood (29.0%) | | Topic 1: Activities in the Neighborhood (22.3%) | | Topic 2: Travel, Penguin Village, and Art (9.5%) | | Topic 3: Local Story and Urban Regeneration (5.2%) | | Topic 4: Café and Vibe (16.8%) | | Topic 5: Famous Places and the Food (17.2%) | |
| Keyword | Importance | Keyword | Importance | Keyword | Importance | Keyword | Importance | Keyword | Importance | Keyword | Importance |
| friend | 0.0102 | thought | 0.0164 | artwork | 0.0107 | human | 0.0087 | café | 0.0730 | famous place | 0.0343 |
| human | 0.0098 | time | 0.0153 | Penguin Village | 0.0096 | missionary | 0.0074 | coffee | 0.0220 | menu | 0.0233 |
| mom | 0.0094 | human | 0.0136 | village | 0.0091 | thanks | 0.0063 | order | 0.0150 | order | 0.0176 |
| unni | 0.0094 | heart | 0.0097 | travel | 0.0089 | church | 0.0062 | dessert | 0.0146 | food | 0.0122 |
| café | 0.0080 | flower | 0.0056 | space | 0.0080 | Korea | 0.0061 | vibe | 0.0122 | recommendation | 0.0112 |
| dinner | 0.0078 | beginning | 0.0055 | area | 0.0074 | life | 0.0047 | famous place | 0.0115 | visit | 0.0096 |
| oppa | 0.0075 | story | 0.0054 | proceeding | 0.0073 | school | 0.0044 | recommendation | 0.0106 | sauce | 0.0090 |

| Theme 1: Daily Life (52.3%) | | | | Theme 3: Travel, Art, and Local Stories (14.7%) | | | | Theme 2: Cafés and Local Hot Spots (34.0%) | | | |
|---|---|---|---|---|---|---|---|---|---|---|---|
| Topic 0: Loved Ones in the Neighborhood (29.0%) | | Topic 1: Activities in the Neighborhood (22.3%) | | Topic 2: Travel, Penguin Village, and Art (9.5%) | | Topic 3: Local Story and Urban Regeneration (5.2%) | | Topic 4: Café and Vibe (16.8%) | | Topic 5: Famous Places and the Food (17.2%) | |
| Keyword | Importance | Keyword | Importance | Keyword | Importance | Keyword | Importance | Keyword | Importance | Keyword | Importance |
| happiness | 0.0051 | book | 0.0052 | afternoon | 0.0071 | history | 0.0042 | beverage | 0.0101 | pasta | 0.0089 |
| morning | 0.0049 | looking around | 0.0046 | introduction | 0.0063 | mother | 0.0041 | space | 0.0100 | vibe | 0.0085 |
| drink | 0.0049 | travel | 0.0045 | exhibition | 0.0063 | name | 0.0038 | bread | 0.0100 | thought | 0.0071 |
| lunch | 0.0047 | love | 0.0043 | street | 0.0058 | love | 0.0034 | spot | 0.0096 | feeling | 0.0069 |
| time | 0.0043 | feeling | 0.0041 | artist | 0.0056 | beginning | 0.0031 | feeling | 0.0091 | friend | 0.0064 |
| today | 0.0042 | child | 0.0041 | visit | 0.0054 | Chosun | 0.0031 | visit | 0.0087 | meat | 0.0061 |
| thought | 0.0040 | preparation | 0.0041 | building | 0.0051 | support | 0.0031 | menu | 0.0085 | pizza | 0.0059 |
| coffee | 0.0039 | name | 0.0037 | culture | 0.0051 | pastor | 0.0031 | interior | 0.0071 | table | 0.0058 |
| love | 0.0039 | thanks | 0.0037 | operation | 0.0050 | country | 0.0030 | friend | 0.0058 | meal | 0.0054 |
| Dongmyu-ngdong | 0.0038 | appearance | 0.0036 | alley | 0.0043 | activity | 0.0030 | thought | 0.0055 | eating place | 0.0053 |
| exercise | 0.0035 | friend | 0.0034 | appearance | 0.0042 | story | 0.0030 | human | 0.0055 | spot | 0.0053 |
| gift | 0.0035 | tea | 0.0034 | city | 0.0041 | exercise | 0.0030 | inside | 0.0051 | Menu (board) | 0.0048 |
| getting off work | 0.0034 | memory | 0.0034 | course | 0.0040 | urban_regeneration | 0.0029 | cake | 0.0051 | soup | 0.0043 |
| feel like | 0.0032 | happiness | 0.0030 | representative | 0.0040 | society | 0.0029 | Leeleenam Studio business owner | 0.0046 | time | 0.0042 |
| dad | 0.0032 | recommendation | 0.0029 | art | 0.0038 | progress | 0.0028 | table | 0.0043 | store | 0.0041 |
| blog | 0.0032 | today | 0.0028 | use | 0.0037 | real estate | 0.0028 | tea | 0.0040 | dish | 0.0036 |
| last | 0.0032 | writing | 0.0028 | place | 0.0037 | earth | 0.0027 | Americano | 0.0040 | cheese | 0.0036 |
| best | 0.0031 | necessity | 0.0025 | participation | 0.0036 | China | 0.0027 | sensibility | 0.0040 | salad | 0.0035 |
| beginning | 0.0030 | arrival | 0.0025 | experience | 0.0035 | America | 0.0027 | latte | 0.0037 | today | 0.0034 |
| daily life | 0.0030 | painting | 0.0024 | site | 0.0034 | Mr. | 0.0026 | time | 0.0035 | human | 0.0034 |
| hair | 0.0029 | gift | 0.0024 | tour | 0.0033 | spirit | 0.0025 | cup | 0.0034 | steak | 0.0033 |
| weather | 0.0029 | feel like | 0.0024 | history | 0.0033 | song | 0.0024 | looking around | 0.0033 | waiting | 0.0033 |
| tea | 0.0029 | one day | 0.0023 | making | 0.0033 | spot | 0.0024 | | 0.0032 | shrimp | 0.0032 |

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
