# Peer review of "Neighborhood Identity Formation and the Changes in an Urban Regeneration Neighborhood in Gwangju, Korea"

_sustainability, doi:10.3390/su151511792_

Round 1
Reviewer 1 Report
Dear Authors
· I encourage the authors to recraft the analyses and the manuscript to balance modelling discussion with LDA
· The topic of this research is both interesting and useful. However, the heavy emphasis on methods, and in some cases fine details without clear purpose to the overall message, reduces the value of this manuscript. I recommend rewriting the manuscript with a new outline based on addressing relevance for new and old business
· You should do better to write the goal of the current research in a sentence at the end of Abstract.
· I suggest that you revise the results section showing the major or most important categories only . It ‘s very confused. The tables presented are too large and the focus is lost .
The discussion does not seem clear regarding the application of these methods in the management . In my opinion, and based on the data presented, this needs further analysis and discussion.
Best regards
Author Response
- I encourage the authors to recraft the analyses and the manuscript to balance modelling discussion with LDA
The authors made some changes marked in red. Please refer to lines 340-344, 369-378, and 549-550 for results and lines 594, 603, and 609-628 for discussion.
- The topic of this research is both interesting and useful. However, the heavy emphasis on methods, and in some cases fine details without clear purpose to the overall message, reduces the value of this manuscript. I recommend rewriting the manuscript with a new outline based on addressing relevance for new and old business.
Because the dataset does not illustrate old businesses, the authors added more arguments on business in the discussion. Please refer to lines 609-628.
- You should do better to write the goal of the current research in a sentence at the end of Abstract.
The authors included one more sentence at the end. Please refer to lines 21-23. The goal of the research was clarified in lines 51-71 (marked in red).
- I suggest that you revise the results section showing the major or most important categories only. It‘s very confused. The tables presented are too large and the focus is lost.
Some revisions were made in the results and some sentences were rearranged for further clarification (lines 340-344, 369-378, 481-482, 549-550). The tables were moved to Appendix.
- The discussion does not seem clear regarding the application of these methods in the management. In my opinion, and based on the data presented, this needs further analysis and discussion.
Authors added more argument in 609-629.
Reviewer 2 Report
Using data mining from websites, the paper analyses the place identity and sentiments of the Yanglim area of Gwangju, South Korea. The methodology is interesting, and the topic is relevant to urban planning and development. I have the following comments to improve the academic clarity and rigor of the paper.
The abstract does not reflect how regeneration is a valid research topic and how it offers broader implications based on the case study. Having one or two rational sentences toward the beginning of the abstract will be great. One to two sentences on broader implications are required.
The writing is unclear, and thorough edits are required. For example: “….the analysis of social media data was analyzed as a newly emerging computational method”.
The writing is not coherent. Some sentences are strange. :” Before 1904, 130 Yanglim was closely located in the city center, but the land around it was affordable, including a great many graveyards.”
Is the first paragraph of the literature review from Lynch’s work? Why did the authors not include the work on these ideas (identity, structure, and meaning) from other sources and case studies? There are many more publications on this topic besides the original work by Lynch. The image of the city has been transformed since the 1960s. Narratives from recent publications are expected.
The second paragraph of lit. The review has the same issue. It does not have enough citations to support the authors’ claims on neighborhood identity. The discussion between social research and GIS approaches is unbalanced in the second paragraph. There are too many citations to support quantitative approaches but not much towards the social methods of perception assessments.
The second part of the literature review is not a review of previous works. Rather it is the overview of the study area, and some of the descriptions are not linked to the study's main objective, which is place identity. For example, it is unclear how the last paragraph is related to the place identity.
The data collection method should be explained in non-technical terms. As it is, I felt that the authors assume that readers are familiar with the data-scouring approach (Crawling). What are the data sources? They must be described.
The analysis method seems to be AI or deep learning approach. There is too much technical jargon without descriptions in section 3.2. What is corpus preparation, MeCab-Ko, LDA, Genism score, etc? It is unclear what the exact method of the paper was and what application the authors used to run the analysis. They should be supported and explained using citations.
Why were the periods of analysis categorized into three? The rationale could be government interventions with different policies and programs. What are the rationales for duration breakdowns? How the distribution of terms is determined across stages. How did the importance of the key terms were determined (Figure 7)?
There are large tables and too many figures in 4.2.1 without complete explanations. The number of figures should be reduced, and tables could be condensed. Table 4 is too large without clear significance and explanations. The same goes for Tables 5 and 6.
Table 7 has just one row. It can be presented and explained in a paragraph.
The writing is unclear, and thorough edits are required. For example: “….the analysis of social media data was analyzed as a newly emerging computational method”.
The writing is not coherent. Some sentences are strange. :” Before 1904, 130 Yanglim was closely located in the city center, but the land around it was affordable, including a great many graveyards.”
Author Response
Using data mining from websites, the paper analyses the place identity and sentiments of the Yanglim area of Gwangju, South Korea. The methodology is interesting, and the topic is relevant to urban planning and development. I have the following comments to improve the academic clarity and rigor of the paper.
- The abstract does not reflect how regeneration is a valid research topic and how it offers broader implications based on the case study. Having one or two rational sentences toward the beginning of the abstract will be great. One to two sentences on broader implications are required.
Please read the first sentence of the abstract marked in red. Lines 8-9.
- The writing is unclear, and thorough edits are required. For example: “….the analysis of social media data was analyzed as a newly emerging computational method”.
The authors appreciate your example. I changed the sentences adding arguments marked in red. Please refer to lines 51-71.
- The writing is not coherent. Some sentences are strange. :” Before 1904, 130 Yanglim was closely located in the city center, but the land around it was affordable, including a great many graveyards.”
The authors revised this sentence. Please refer to lines 147-149.
- Is the first paragraph of the literature review from Lynch’s work? Why did the authors not include the work on these ideas (identity, structure, and meaning) from other sources and case studies? There are many more publications on this topic besides the original work by Lynch. The image of the city has been transformed since the 1960s. Narratives from recent publications are expected.
The authors rewrote section 2.1 switching the order of sentences and adding more literature review. This part is marked in red.
- The second paragraph of lit. The review has the same issue. It does not have enough citations to support the authors’ claims on neighborhood identity. The discussion between social research and GIS approaches is unbalanced in the second paragraph. There are too many citations to support quantitative approaches but not much towards the social methods of perception assessments.
The authors rewrote section 2.1 switching the order of sentences and adding more literature review. This part is marked in red.
- The second part of the literature review is not a review of previous works. Rather it is the overview of the study area, and some of the descriptions are not linked to the study's main objective, which is place identity. For example, it is unclear how the last paragraph is related to the place identity.
The authors rewrote section 2.1 switching the order of sentences and adding more literature review for the overall flow and argument. This part is marked in red.
- The data collection method should be explained in non-technical terms. As it is, I felt that the authors assume that readers are familiar with the data-scouring approach (Crawling). What are the data sources? They must be described.
Web Crawling is similar to Web Scraping but the term, Web Scraping is more appropriate for this study. The term web scraping was explained in lines 214-217. The data source is described in lines 217-231.
- The analysis method seems to be AI or deep learning approach. There is too much technical jargon without descriptions in section 3.2. What is corpus preparation, MeCab-Ko, LDA, Genism score, etc? It is unclear what the exact method of the paper was and what application the authors used to run the analysis. They should be supported and explained using citations.
These terms were explained more in Sec. 3.2. Refer to lines 233-244 and 247-248. Marked in red. The term genism was erased to avoid too much technical explanation.
9. Why were the periods of analysis categorized into three? The rationale could be government interventions with different policies and programs. What are the rationales for duration breakdowns?
The authors clarified the goal of this research, arguing why using technology is significant (Lines 51-71). To answer the research questions, the period was categorized by trend analysis of the data in Section 3.3 (lines 369-378).
How the distribution of terms is determined across stages.
The clarification is written in lines 300-304 and 333-344.
How did the importance of the key terms were determined (Figure 7)?
Lines 289-291 marked in red.
- There are large tables and too many figures in 4.2.1 without complete explanations. The number of figures should be reduced, and tables could be condensed. Table 4 is too large without clear significance and explanations. The same goes for Tables 5 and 6.
Tables 3, 4, 5, & 6 are inserted in Appendix.
- Table 7 has just one row. It can be presented and explained in a paragraph.
More information is added in lines 609-628.
Comments on the Quality of English Language
- The writing is unclear, and thorough edits are required. For example: “….the analysis of social media data was analyzed as a newly emerging computational method”.
It is the same comment as Q2. Please refer to Q2.
- The writing is not coherent. Some sentences are strange. :” Before 1904, 130 Yanglim was closely located in the city center, but the land around it was affordable, including a great many graveyards.”
It is the same comment as Q3. Please refer to Q3.
Reviewer 3 Report
The study addresses the topical issue of urban environmental changes due to the covid-19 pandemic and urban identity formation. Text is well written and provides an interesting perspective on the issue for potential discussions among experts in the field. However, I would like to make some remarks:
I would like to add that (row 66) "An environmental image may possess three components: identity, structure, and meaning..." the image of a place is formed to a much greater extent by the subjective perceptions of individuals, which are even more important than objective reality (ref. Matlovicova, K and Kormanikova, J. 2014. City Brand-Image Associations Detection. Case Study of Prague. International Multidisciplinary Scientific Conferences on Social Sciences and Arts (SGEM 2014).Psychology and psychiatry, sociology and healthcare, education, VOL II , pp.139-146).
For this reason, it is necessary to distinguish between the image and the identity of the territory, which shape the overall reputation of the territory. These approaches are then reflected in the planning policies of tourist destinations development, which reflect the specific character of the cultural heritage of the territory (ref. Matlovicova, K and Husarova, M. 2017. Potential of the Heritage Marketing in Tourist Destinations Development. Cicva castle ruins case study. Folia Geographica 59 (1) , pp.5-35).
Even more important is the subjective perception in the case of changes in the identity of the territory, when it is necessary to reflect possible deviations in destination marketing. The formation of a place identity is directly dependent on perceptions. There is a mutual implication between identity - image and reputation. This is best illustrated by the perception of security in an urban area. There are studies that confirm this interconnected relationship (ref. Matlovicova, K; Mocak, P and Kolesarova, J. 2016. Environment of estates and crime prevention through urban environment formation and modification. Geographica Pannonica 20 (3) , pp.168-180).
In conclusion, it is a quality study based on a well-developed and original methodology for obtaining and processing relevant data. The paper has a logical structure, relies on relevant sources and provides an interesting perspective on the undoubtedly widely discussed problem the urban identity changes after Covid-19 pandemic.
However, the above comments in no way diminish the quality of the study. It is balanced in content, uses correct methods and I definitely recommend it for publication after minor changes.
Author Response
The study addresses the topical issue of urban environmental changes due to the covid-19 pandemic and urban identity formation. Text is well written and provides an interesting perspective on the issue for potential discussions among experts in the field. However, I would like to make some remarks:
- I would like to add that (row 66) "An environmental image may possess three components: identity, structure, and meaning..." the image of a place is formed to a much greater extent by the subjective perceptions of individuals, which are even more important than objective reality (ref. Matlovicova, K and Kormanikova, J. 2014. City Brand-Image Associations Detection. Case Study of Prague. International Multidisciplinary Scientific Conferences on Social Sciences and Arts (SGEM 2014). Psychology and psychiatry, sociology and healthcare, education, VOL II , pp.139-146).
The authors added the citation you mentioned. Please refer to lines 83-85.
- For this reason, it is necessary to distinguish between the image and the identity of the territory, which shape the overall reputation of the territory. These approaches are then reflected in the planning policies of tourist destinations development, which reflect the specific character of the cultural heritage of the territory (ref. Matlovicova, K and Husarova, M. 2017. Potential of the Heritage Marketing in Tourist Destinations Development. Cicva castle ruins case study. Folia Geographica 59 (1) , pp.5-35).
Thank you for recommending this paper for a citation but I could not cite this since it’s beyond my understanding (not written in English).
- Even more important is the subjective perception in the case of changes in the identity of the territory, when it is necessary to reflect possible deviations in destination marketing. The formation of a place identity is directly dependent on perceptions. There is a mutual implication between identity - image and reputation. This is best illustrated by the perception of security in an urban area. There are studies that confirm this interconnected relationship (ref. Matlovicova, K; Mocak, P and Kolesarova, J. 2016. Environment of estates and crime prevention through urban environment formation and modification. Geographica Pannonica 20 (3) , pp.168-180).
The authors also cited Matlovicova et. al’s work (2016). Please refer to lines 83-85.
In conclusion, it is a quality study based on a well-developed and original methodology for obtaining and processing relevant data. The paper has a logical structure, relies on relevant sources and provides an interesting perspective on the undoubtedly widely discussed problem the urban identity changes after Covid-19 pandemic.
However, the above comments in no way diminish the quality of the study. It is balanced in content, uses correct methods and I definitely recommend it for publication after minor changes.
Round 2
Reviewer 1 Report
Dear Authors
The study is very interesting .
I suggest that the authors review their results. They are still very confusing, not easily readable, and should focus on a few main categories.
On the other hand, the multiple figures referenced as 10 also become confusing for the reader.
best regards
Author Response
The study is very interesting.
I suggest that the authors review their results. They are still very confusing, not easily readable, and should focus on a few main categories.
- The authors tried to clarify the results based on themes or topics. A table moved to Appendix A. All revisions are marked in red.
On the other hand, the multiple figures referenced as 10 also become confusing for the reader.
- The authors wrote a rationale for the word clouds in Lines 289-291.
best regards
- English was edited by a professional editor three times in total.
